# SpikingVTG: Saliency Feedback Gating Enabled Spiking Video Temporal Grounding

## Abstract

Video Temporal Grounding (VTG) seeks to retrieve consecutive intervals or specific clips from a video based on specified natural language queries. VTG requires accurately aligning video segments with corresponding natural language instructions, highlighting the need for effective methodologies to capture semantic correspondence and maintain temporal coherence. Spiking neural networks (SNNs), previously underexplored in this domain, present a unique opportunity to tackle VTG challenges from both the architectural and energy-efficiency perspectives. In this paper, we leverage sparse spike-based communication of SNNs to propose a multimodal architecture tailored for VTG tasks, namely SpikingVTG, providing a biologically inspired and efficient solution. Leveraging temporal saliency feedback, our proposed spiking video-language model (VLM) achieves competitive performance with non-spiking VLMs across diverse moment retrieval and highlight detection tasks. We introduce a Saliency Feedback Gating (SFG) mechanism that improves performance while reducing overall neural activity. To efficiently train our spiking VLM, we analyze the convergence dynamics of each neuronal layer and utilize equilibrium states to enable training using implicit differentiation at equilibrium. This approach eliminates the need for computationally expensive backpropagation through time while also enabling the use of knowledge distillation for efficient model training. To further improve operational efficiency and facilitate the on-chip deployability of our model, we leverage a multi-stage training pipeline that focuses on eliminating non-local computations, such as softmax and layer normalization, leading to the development of the Normalization Free (NF)-SpikingVTG model. Additionally, we create an extremely quantized variant, a 1-bit NF-SpikingVTG model, which vastly improves computational efficiency during inference while maintaining minimal performance degradation from our base model. Our work introduces the first spiking model to demonstrate competitive performance on VTG benchmarks, including QVHighlights and Charades-STA.

## 1 Introduction

The rapid expansion of various social medias and portable smart technologies has triggered an unprecedented surge in video content. This vast influx of data has intensified the need for efficient methods to retrieve and analyze video information. Consequently, the field of Video Temporal Grounding (VTG) (Lei et al., 2021; Lin et al., 2023) has emerged as an important area of research. The main objective of VTG is to identify the precise segment of a video that corresponds to a given natural language query, enabling accurate and context-driven video content retrieval. In this paper, we focus on two tasks: moment retrieval (Zhang et al., 2020; Mun et al., 2020), which aims to identify video intervals relevant to a given query, and highlight detection (Hong et al., 2020), which retrieves the best candidate segment of the video in response to the query. Our work involves analyzing multimodal data—combining video content with natural language queries—to develop an effective solution to the problem. With the rise of foundation models like large language models (LLMs) and video-language models (VLM), the field of VTG has seen significant advancements (Liu et al., 2022; Lei et al., 2021). However, these models demand substantial computational power and energy (Samsi et al., 2023) to operate. Furthermore, VTG is inherently resource-intensive, requiring the analysis of long video sequences, leading to significant computational overhead. In this work, we leverage sparse spike-based communication and simplified accumulation-based compu-

tation in spiking neural networks (SNNs) (Ghosh-Dastidar & Adeli, 2009) to develop an efficient, lightweight solution for VTG.

Beyond the computational efficiency of SNN-based frameworks, we also harness their temporal dynamics to propose a spiking transformer-based VLM (Fig. 1), namely **SpikingVTG**, that matches or surpasses the performance of current state-of-the-art non-spiking VLM. The input video for the VTG task typically consists of a long sequence of segments or clips. A key challenge in VTG is thus accurately identifying salient segments (Lin et al., 2023) or temporally dependent segments that exhibit a strong semantic correspondence with the given query. Our SNN-based VLM, operated over a period of simulation time steps, allows us to leverage its intermediate temporal output as a feedback to identify the salient segments. We use the average spiking rate (ASR) of the output of the spiking transformer core in the SpikingVTG model to compute a dynamic saliency score of each video segment w.r.t the given query, which we then leverage as a mask for a multiplicative gating mechanism. This improves performance of the model by enabling it to focus on relevant portions of the video, while also reducing computational overhead by minimizing attention to irrelevant segments.

From a bio-plausibility perspective, as explored by Kar et al. (2019), feedback based connection plays a prominant role in human visual cortex primarily responsible for object recognition. Furthermore, the feedback connection maintains the layer-wise convergence of ASR at equilibrium, enabling the implementation of an implicit differentiation framework (Xiao et al., 2021), allowing for more efficient training of our model. This learning framework, leverages layer-wise converged ASR values at equilibrium to train the spiking model in one backpropagation step, instead of using the computationally expensive backpropagation through time (BPTT) (Neftci et al., 2019). The SpikingVTG framework further involves a multi-stage training pipeline aimed at developing spiking models to facilitate potential deployment on resource-constrained edge-based device enabled with neuromorphic chips. To allow for efficient training of our spiking model, we employ a knowledge distillation strategy (Hinton et al., 2015), enabling knowledge transfer from a non-spiking UniVTG model, used as the "teacher", to our "student" SpikingVTG model. This process utilizes the ASR of converged intermediate states at equilibrium, enabling efficient training of our spiking VLM.

Traditional transformer architectures (Vaswani et al., 2017) utilize non-local normalization operations such as softmax and layer normalization, which present challenges for implementation on neuromorphic hardware (Shrestha et al., 2022). To address this limitation, we introduce the Normalization-Free (NF)-SpikingVTG model, which eliminates all layer normalization operations and substitutes softmax spiking attention with a ReLU-based spiking attention mechanism. Although, Softmax-free attention has been explored in the literature (Koohpayegani & Pirsiavash, 2024; Xu et al., 2024), it has predominantly been applied to vision tasks. While, ReLU-based attention mechanisms have previously been explored in non-spiking domains (Shen et al., 2023), we are the first to introduce this concept within a spiking attention mechanism. Additionally, to reduce computational complexity, following works on quantization in analog LLMs (Wang et al., 2023), we propose a 1-bit quantized variant of SpikingVTG. Our multi-stage training pipeline enables minimal performance degradation while enhancing computational efficiency during inference, in our SpikingVTG models. To our knowledge, this work is the first to evaluate an operational spiking VLM framework across various VTG tasks, including moment retrieval and highlight detection, on datasets such as QVHighlights and Charades-STA.

The primary contributions of our work are as follows:

- **SpikingVTG Model and Training Framework:** We propose a transformer-based, multimodal spiking video language model with a spiking decoder module for moment retrieval and highlight detection in VTG tasks. We leverage the layer-wise convergence dynamics in our model to train our model using implicit differentiation at equilibrium, bypassing memory intensive BPTT. The result is the first spiking architecture to demonstrate competitive performance on VTG.

- **Saliency Feedback Gating Mechanism:** We introduce a saliency feedback gating mechanism for input video, that leverages the ASR of the output of the spiking transformer core at each time step. This temporal feedback enhances task-specific performance while minimizing neural activity, ultimately reducing overall computational overhead.

- **Multi-Stage Training Pipeline:** We propose a multi-stage training pipeline for our SpikingVTG framework, utilizing knowledge distillation and architectural modifications to cre-

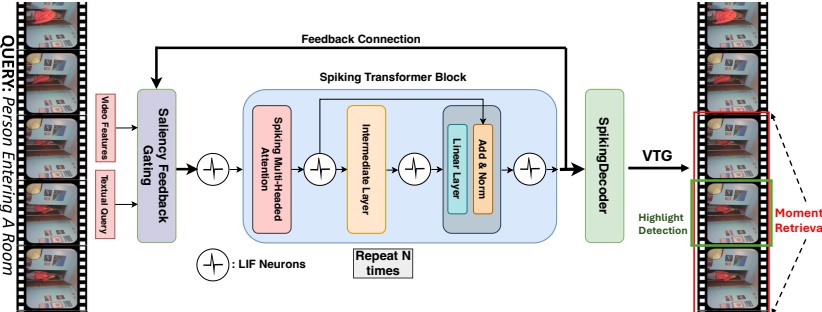

Figure 1: High-level overview of the proposed SpikingVTG architecture. The spiking Vision-Language Model (VLM) takes video and textual features as inputs, employing a spiking transformer core that utilizes Saliency Feedback Gating through temporal feedback connections. The model incorporates a spiking decoder module that takes the output of the transformer core to predict parameters for the VTG task.

ate lightweight and computationally efficient spikingVTG variants. We replace computationally intensive non-local operations like layer normalization and softmax with hardware-friendly alternatives. We further introduce extreme quantization, developing a 1-bit NF-SpikingVTG model that significantly reduces memory as well as computational overhead.

## 2 RELATED WORKS

**VTG Advancements:** With the recent rise of multimodal LLM architectures, the field of video-language modeling has opened new avenues for understanding and extracting key information from video data. Moment-DETR (Lei et al., 2021), a transformer encoder-decoder model introduced alongside the QVHighlights dataset, laid a strong foundation for subsequent VTG architectures. UMT (Liu et al., 2022) introduced an unified framework for solving both highlight detection and moment retrieval tasks. Due to the limited availability of trainable video data, UniVTG (Lin et al., 2023) proposed an innovative solution by unifying various VTG tasks and labels under a single formulation. This approach enabled the development of an LLM-like pretraining framework, achieving state-of-the-art performance on VTG tasks. Although no fully spiking-based architecture has been explored for VTG tasks, SpikeMba (Li et al., 2024)—primarily a non-spiking model—integrates SNN components to generate proposal sets from video data. However, since its core framework is derived from Mamba (Gu & Dao, 2023) and relies on floating-point matrix multiplications, SpikeMba cannot be considered a baseline for spiking models, which predominantly use accumulation-based operations.

**Spiking neural networks (SNNs):** SNNs allow for event-driven computation and communication in neuromorphic hardware, significantly reducing energy consumption. SNNs have been implemented in neuromorphic systems like IBM TrueNorth (DeBole et al., 2019) and Intel Loihi 2 (Davies et al., 2021), demonstrating approximately $75\times$ greater energy efficiency compared to traditional neural networks running on low-power GPUs (Tang et al., 2020). SNNs, with their energy-efficient computational framework, offer a promising solution to the resource-intensive demands of multimodal VTG tasks. While SNNs for a long time were confined in simpler vision-based tasks (Yamazaki et al., 2022) with relatively simple architectures, recent developments have scaled them to transformer-based architectures for tasks ranging from vision to language modelling (Zhou et al., 2022; Bal & Sengupta, 2024; Zhu et al., 2023), however majority of them rely on some normalization techniques which are not implementable on a neuromorphic chip.

## 3 METHODOLOGY

In this section, we first present the VTG problem formulation, followed by a detailed explanation of the SpikingVTG framework. We describe its core components, including the spiking transformer, saliency feedback gating mechanism, and spiking decoder. Next, we elaborate on the scalable train-

ing framework and then we introduce the multi-stage pipeline, that allows efficient training using knowledge distillation and enables more efficient iterations of our architecture, facilitating the development of lightweight SpikingVTG variants such as NF-SpikingVTG and 1-bit NF-SpikingVTG.

## 3.1 VIDEO TEMPORAL GROUNDING (VTG)

For a given video $V$ and language query $Q$, we start by segmenting $V$ into a sequence of $L_v$ fixed-length clips, denoted as $\{v_1, \ldots, v_{L_v}\}$. Each clip $v_i$ has a length $l$ and is centered at timestamp $t_i$. The textual query $Q$ consists of $L_q$ tokens, denoted as $Q = \{q_1, \ldots, q_{L_q}\}$. Following previous studies on VTG (Lin et al., 2023), we define three parameters for each clip $v_i = (f_i, d_i, s_i)$, where $f_i = 1$ if the clip is in foreground, i.e. relevant else $f_i = 0$. $d_i = [d_{s_i}, d_{e_i}] \in \mathbb{R}^2$ represent the temporal distance that converts the clip timestamp $t_i$ to its interval boundaries. Here, $d_i$ is valid when $f_i = 1$. The term $d_{s_i}$ denotes the distance between the start of the interval and $t_i$, while $d_{e_i}$ denotes the distance between the end of the interval and $t_i$. $s_i \in [0, 1]$ is a continuous score that quantifies the relevance between the visual content of clip $v_i$ and the query $Q$. Our proposed SpikingVTG model predicts these three parameters for each video clip. In this paper, we focus on specific VTG tasks, which are carried out as follows:

**Moment Retrieval:** We rank the predicted clip boundaries $\{\tilde{b}_i\}_{i=1}^{L_v}$, where $b_i = [t_i - d_{s_i}, t_i + d_{e_i}]$, based on their associated probabilities given by $\{\tilde{f}_i\}_{i=1}^{L_v}$. Since the predicted $L_v$ boundaries are dense, we employ a 1-dimensional Non-Maximum Suppression (NMS) (Hosang et al., 2017) with a threshold of 0.7 to eliminate highly overlapping boundary boxes, resulting in a final prediction.

**Highlight Detection** For each clip, we rank all clips based on their combined scores $\{\tilde{f}_i + \tilde{s}_i\}_{i=1}^{L_v}$. This combined value represents how well the chip $i$ match with the underlying query. We then return the top clips (e.g., Top-1) as predictions.

## 3.2 SPIKINGVTG: ARCHITECTURE OVERVIEW

The core computational unit of the proposed SpikingVTG model is a leaky integrate-and-fire (LIF) neuron (Dutta et al., 2017). Neurons communicate with each other using sparse, spike-based activations instead of real-valued signals, significantly improving energy/power efficiency. The model architecture includes a spiking transformer core for processing inputs, a saliency feedback gating mechanism for dynamic input control, and a spiking decoder module to predict the parameters required for the VTG task, as described in Section 3.1.

### 3.2.1 SPIKING NEURAL NETWORKS

The discrete time dynamics of an LIF-based spiking neuron can be given as follows,

$$\begin{aligned} u_i[t + \delta] &= \gamma u_i[t] + W_{(i-1)}(s_{(i-1)}[t]) + b_i, \\ u_i[t + 1] &= u_i[t + \delta] - V_{th_i} s_i[t + 1], \end{aligned} \tag{1}$$

where, at time $t$, $u_i[t]$ is the membrane potential of the $i^{th}$ neuronal layer; $b_i$ indicates a bias term and $\gamma$ is the leaky term. $W_{(i-1)}$ represents the layer-specific operation; $t + \delta$ is an intermediate time step to determine if the neuron fired; $V_{th_i}$ is the threshold of layer $i$. We use a ternary spiking model (Guo et al., 2024) in our work for spike ($s[t + 1]$) generation, thus the spiking operation is given as,

$$s_i[t + 1] = \begin{cases} +1 & \text{if } u_i[t + \delta] \geq V_{th_i}, \\ -1 & \text{if } u_i[t + \delta] \leq -V_{th_i}, \\ 0 & \text{otherwise} \end{cases} \tag{2}$$

This approach enhances performance while avoiding the introduction of additional floating-point multiplicative and accumulative (fp-MAC) operations. The average spiking rate (ASR) (Xiao et al., 2021) of LIF neurons within each layer $i$ at time $t$ can be defined as a weighted-average function:

$$a_i[t] = \frac{\sum_{\tau=1}^{t} \gamma^{t-\tau} s_i[\tau]}{\sum_{\tau=1}^{t} \gamma^{t-\tau}}. \tag{3}$$

### 3.2.2 SPIKING TRANSFORMER CORE

The high-level overview of each encoder block of our spiking transformer architecture is demonstrated in Fig. 1. The model consists of $N$ encoder layers, each consists of a spiking multi-headed attention block, followed by an intermediate layer and an output layer. Communication within and between encoder layers occurs via spikes. Furthermore, all matrix multiplications involved in linear layers and attention layer comprises of more efficient fp-accumulative (ACC) operations instead of fp-MAC operations in conventional neural architectures. For this work we have used four encoder layers and eight attention heads. The hidden size dimension is 1024. Detailed descriptions of each layer are provided in the Appendix A.1. In Section 3.4.2, we replace non-local normalization operations and introduce a ReLU-based attention mechanism. In Section 3.4.3, we quantize all weights in the linear layers, including those in the intermediate and output layers, to 1-bit precision.

### 3.2.3 SALIENCY FEEDBACK GATING (SFG)

SpikingVTG operates over a specific number of convergence time steps ($T$), with the convergence dynamics detailed in Section 3.3. This temporal processing allows us to leverage intermediate temporal outputs to dynamically update the input to the model at every time step for better predictions. This approach conforms to the feedback connections observed in the human visual cortex (Kar et al., 2019), providing a bioplausible explanation for its efficacy. The ASR of the final encoder layer of the Spiking Transformer core is used as a temporal feedback to compute a dynamic saliency score with the input query enabling the design of a gating mechanism, allowing selective focusing on relevant segments of the video while minimizing computation on irrelevant segments. The saliency feedback gating mechanism is shown below,

Figure 2: Overview of the internal operations of the saliency-feedback gating mechanism. The ASR of the output of the spiking transformer core at each time step is leveraged as the feedback signal (Fig. 1).

$$
\begin{aligned}
F_s^{v_i}[t] = \cos(\mathbf{a_i^{N_v}}[\mathbf{t}], \mathbf{M}) &:= \frac{\mathbf{a_i^{N_v}}[\mathbf{t}] \cdot \mathbf{M}}{\|\mathbf{a_i^{N_v}}[\mathbf{t}]\|_2 \|\mathbf{M}\|_2}, \\
\bar{V}[t+1] &= V * F_s^v[t], \\
I[t+1] &= \bar{V}[t+1] \oplus \mathbf{Q},
\end{aligned}
\tag{4}
$$

where, using attentive pooling operation, sentence representation $\mathbf{M} = \mathbf{Q}^T Softmax(\mathbf{Q}\mathbf{W_p})$, $\mathbf{M} \in \mathbb{R}^D$, textual query features $\mathbf{Q} \in \mathbb{R}^{L_q \times D}$, input video features $V \in \mathbb{R}^{L_v \times D}$ and $\mathbf{W_p} \in \mathbb{R}^{D \times 1}$ is a learnable embedding. $F_s^{v_i}[t]$ is the dynamic saliency score, at time $t$, for the $i$-th segment of the video. The ASR of the output of the spiking transformer core is given as $a^N[t] \in \mathbb{R}^{(L_v + L_q) \times D}$. $a_i^{N_v}[t]$ is ASR of output of the spiking transformer core, corresponding to video segment $i$, at time $t$. The SFG layer comprises $O(L_v \cdot D)$ floating-point multiplication operations; however, the computational overhead of this layer is significantly less than that of the more substantial transformer component which has a complexity of $O(L^2 \cdot D + L \cdot D^2)$, where $L = L_v + L_q$. and $D$ is the hidden dimension of the transformer. $I[t+1]$ is derived from the concatenation of saliency feedback gated video features and query features and serves as the input to the spiking transformer core at time $t+1$.

The SFG mechanism not only results in better performance of our SpikingVTG architecture on evaluation metrics (see Table 3) but also reduces overall neural activity by sparsifying input spikes. As shown in Fig. 3b, empirical results confirm that the model with the gating mechanism exhibits a lower neural activity, particularly in the input and spiking attention layers, compared to the model without this mechanism.

### 3.2.4 SPIKING DECODER

The spiking decoder comprises of stacked 1-D convolutions followed by integrate-fire (IF) neuron layers ($\gamma = 1$ in Eqn. 1), for spike generation. The spiking decoder used for predicting foreground indicator ($f_i$) per clip, applies $n_1$ 1-D convolution operations with kernel size $k_1$, each followed

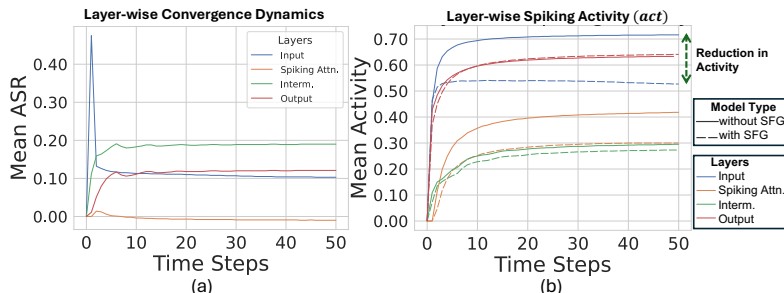

Figure 3: Results obtained upon passing a random input sample from QVHighlights dataset to our SpikingVTG models. (a) Graph shows convergence dynamics of layer-wise mean ASR against operating time steps for a randomly selected spiking transformer encoder layer (Fig. 1). It is to be noted that since we allow ternary spikes ASR can be negative as well. (b) Graph shows, layer-wise mean spiking activity ($act_i[t]$, averaged over number of neurons in that layer) against operating time steps in x-axis. The model with SFG shows markedly reduced activity in both the input layer and the spiking attention layer, underscoring its role in minimizing neuronal activity.

by an IF layer. The final layer consists of a single output channel, and its temporal mean is passed through a sigmoid activation to produce the prediction. The spiking decoder used for $d_i$ applies $n_2$ 1-D convolution operations with kernel size $k_2$, each followed by an IF layer, and the final convolution layer has two output channels to predict $d_i = [d_{s_i}, d_{e_i}]$, after which we compute $b_i$.

## 3.3 TRAINING LEVERAGING CONVERGENCE DYNAMICS

Following, Eqn. 1 & 3, we can formulate $a_i[t+1] = \frac{1}{V_{th_i}}(\hat{f}(a_{(i-1)}[t+1]) + b_i - \frac{u_i[t+1]}{\sum_{j=0}^{t} \gamma^j})$, where $\hat{f}$ is operation of layer $i$. As time approaches $t \to \infty$, the layer-wise ASRs converge to equilibrium, enabling the derivation of steady-state equations for linear layers (Xiao et al., 2021). Moreover, surrogate steady-state functions can be formulated for non-linear layers (Bal & Sengupta, 2024) as,

$$a_i^* = \sigma(\frac{1}{V_{th_i}}(\hat{f}(a_{i-1}^*) + b_i)) \tag{5}$$

where, clipping function $\sigma(x)$ clamps the values within $[-1, 1]$. This is because following Eqn. 2, we allow ternary spikes thus ASR must be with $[-1, 1]$. The empirical convergence of ASR is shown in Fig. 3a. To analyze the overall layer-wise neural activity, which includes both positive and negative spiking event, we present the layer-wise dynamics of the absolute spiking events in Fig. 3b, i.e. $act_i[t] = \frac{\sum_{i=1}^{t} |s_i[t]|}{t}$.

**Training:** As described in the Section 3.2.4, the SpikingDecoder is responsible for predicting $\tilde{f}_i$ and $\tilde{d}_i$ for individual video clip $i$ and $\tilde{s}_i$ is computed using the SFG module at equilibrium. Using these three predictions, we design a loss function that combines various components. The total loss over $N$ clips in the training set is given by $L = \frac{1}{N} \sum_{i=1}^{N} (L_{f_i} + L_{d_i} + L_{c_i})$, where $L_f$ is the binary cross-entropy loss for the indicator variable $f_i$, $L_d$ combines smooth L1 loss with generalized IoU loss (Rezatofighi et al., 2019) for the predicted boundaries, and $L_c$ is an optional loss incorporating intra- and inter-video contranstive learning (Chen et al., 2020). A detailed mathematical formulation of the loss functions can be found in the Appendix B.

During training, leveraging implicit differentiation (Bai et al., 2019) at equilibrium, only ASR values at equilibrium are used,

$$\frac{\partial L(a^*)}{\partial \theta} = -\frac{\partial L(a^*)}{\partial a^*}(J_{g_\theta}^{-1}|_{a^*})\frac{\partial f_\theta(a^*)}{\partial \theta}, \tag{6}$$

where, $\theta$ is the model parameters, $g_\theta(a) = f_\theta(a) - a$, $f$ is the steady-state equation of ASR, $J^{-1}$ is the inverse Jacobian of $g_\theta$ when $a = a^*$, i.e., at equilibrium. Thus, unlike BPTT, we do not need to store the intermediate computational graph and the model parameters can be updated using a single backpropagation step.

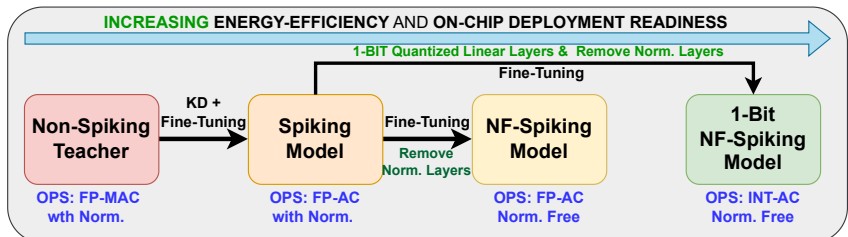

Figure 4: High-level overview of the multi-stage training framework for our proposed SpikingVTG models, enabling the development of lightweight and computationally efficient spiking models. Below each model we have noted the primary operations involved in that architecture.

### 3.4 Multi-Staged Training Pipeline

Training a multimodal spiking architecture like SpikingVTG is resource-intensive. To enhance the efficiency of this process and develop computationally efficient variants of our model, we propose a multi-staged training framework, as illustrated in Fig. 4. We utilize a non-spiking "teacher" VLM to guide the training of our "student" SpikingVTG model. After this initial stage, we fine-tune SpikingVTG using the true labels. Once the base SpikingVTG model is established, we modify its architecture, as outlined in Sections 3.4.2 and 3.4.3, followed by additional fine-tuning to create computationally efficient variants with minimal performance degradation. The resulting computationally efficient, lightweight models are well-suited for deployment on neuromorphic chips, enabling efficient inference.

### 3.4.1 Leveraging Knowledge Distillation (KD)

To enable efficient training of our spiking multimodal architecture, we utilize Knowledge Distillation (KD) techniques (Hinton et al., 2015; Tang et al., 2019; Jiao et al., 2019). We use the pre-trained UniVTG, currently the state-of-the-art in VTG, as a "teacher" and rather than distilling based on the prediction layer, we exploit the outputs of the internal layers of the "teacher". We establish a one-to-one mapping, by design, between the internal representations at equilibrium of our spiking transformer and the corresponding layers of the "teacher", ensuring that the number of layers in both architectures remains consistent. The internal representation-based KD is formulated as,

$$L_{KD} = \sum_{i=1}^{N} MSE(a_{r_i}^* W_d, T_{r_i}) \tag{7}$$

where, $W_d \in \mathbb{R}^{d_s \times d_t}$ is a linear transformation that aligns the dimensionality of the layers of the "student" with that of the corresponding layers of the "teacher". $a_{r_i}^*$ denotes the converged ASR at equilibrium of the internal representation layer $r_i$, which is the output from the spiking transformer encoder layer $i$ of the "student". $T_{r_i}$ is the representation of the the corresponding block $i$ of the teacher model. The KD process is an integral part of the framework and serves as the first stage of our multi-stage pipeline, followed by fine-tuning on the true labels (Fig. 4).

### 3.4.2 Replacing Softmax and Removing Layer Normalization

In our work, we use a spiking attention mechanism (see Appendix A.1) which uses the key and value inputs as spikes instead of real values. Given $d$-dimensional queries, keys, and values $\{q_i[t], s_{k_i}[t], s_{v_i}[t]\}_{i=1}^{L}$, at time $t$, the attention weights $\alpha_{ij}$ are computed as follows:

$$\alpha_{ij}[t] = \phi\left(\frac{1}{\sqrt{d}}\left[q_i[t]^\top s_{k_1}[t], \cdots, q_i[t]^\top s_{k_L}[t]\right]\right)_j \tag{8}$$

where, $\phi$ is the softmax function and output of spiking attention layer at time $t$ is $attn_i[t] = \sum_{j=1}^{L} \alpha_{ij}[t]s_{v_j}[t]$. Given that softmax requires expensive non-local fp-MAC operations, we replace

it with the less costly ReLU() operation and perform a simple scaling with $L^{-1}$. This replacement, while maintaining competitive model performance, is only feasible when following the multi-stage training process outlined in Fig. 4. This highlights the importance of the initial KD and fine-tuning stages, which help stabilize the model. Additionally, we explore the removal of all layer normalization layers, from Fig. 1, during training (as shown in Fig. 4), further streamlining the model design for on-chip deployment. We refer to the resulting model, which uses ReLU in place of Softmax and omits layer normalization, as a Normalization-Free (NF) spiking model.

### 3.4.3 1-BIT WEIGHT QUANTIZED SPIKINGVTG

Following prior work (Wang et al., 2023), 1-bit quantization consists of centering the weights $W$ to achieve a zero mean, followed by binarization to $+1$ or $-1$ using the signum function as shown,

$$
W_q = \text{sgn}(W - \alpha),
$$
$$
\alpha = \frac{1}{nm} \sum_{ij} W_{ij} \tag{9}
$$

where, $W \in \mathbb{R}^{n \times m}$. The signum function, denoted as $\text{sgn}(x)$, categorizes the element $x$ based on its sign. It outputs $+1$ when $x$ is positive and $-1$ when $x$ is zero or negative. The output of the linear layer is scaled by a constant $\beta = \frac{1}{nm} \sum_{ij} |W_{ij}|$. Thus, with ternary activations, our model now incorporates binary weights. Following the multi-stage learning approach illustrated in Fig. 4, our 1-bit SpikingVTG model emerges as a light-weight multimodal spiking VLM, with all linear layer weights quantized to 1-bit. Additionally, we empirically demonstrate that employing binary weights while eliminating normalization layers achieves competitive performance, resulting in 1-bit NF-SpikingVTG, enabling on-chip implementation and significantly improving computational efficiency. Thus, in the resulting model the primary computational operation involve integer accumulations since individual weight values are $W_{q_{ij}} \in \{-1, 1\}$ and activations are $s \in \{-1, 0, 1\}$.

Table 1: Performance comparison of our SpikingVTG model with SFG against non-spiking VTG solutions on the evaluation set of the QVHighlights and Charades-STA for **moment retrieval task**.

| Method | SNN | QVHighlights | | | | Charades-STA | | | |
|---|---|---|---|---|---|---|---|---|---|
| | | @0.3 | @0.5 | @0.7 | mAP@avg | @0.3 | @0.5 | @0.7 | mIoU |
| UniVTG+PT (Lin et al., 2023) | No | 78.58 | 67.35 | 52.65 | 45.44 | 72.63 | 60.19 | 38.55 | 52.17 |
| M-DETR (Lei et al., 2021) | No | - | 53.94 | 34.84 | 32.20 | 65.83 | 52.07 | 30.59 | 45.54 |
| 2D-TAN (Zhang et al., 2020) | No | - | - | - | - | 58.76 | 46.02 | 27.5 | 41.25 |
| LLaViLo (Ma et al., 2023) | No | - | - | - | - | 55.72 | 33.43 | - |
| UniVTG (Lin et al., 2023) | No | 71.81 | 59.74 | 40.90 | 36.13 | 70.81 | 58.01 | 35.65 | 50.1 |
| UMT (Liu et al., 2022) | No | - | 60.26 | 44.26 | 38.59 | - | 49.35 | 26.16 | - |
| EaTR (Jang et al., 2023) | No | - | 61.36 | 45.79 | 41.74 | - | - | - | - |
| QD-DETR (Moon et al., 2023) | No | - | 62.68 | 46.66 | 41.22 | - | 57.31 | 32.55 | - |
| EMTM (Liang et al., 2023) | No | - | - | - | - | 72.70 | 57.91 | 39.80 | 53.00 |
| $R^2$ - Tuning (Liu et al., 2024) | No | - | 68.03 | 49.35 | 46.17 | 70.91 | 59.78 | 37.02 | 50.86 |
| SpikeMba (Li et al., 2024) | No | - | 65.32 | 51.33 | 44.84 | 71.24 | 59.65 | 36.12 | 51.74 |
| **SpikingVTG (Our Model)** | **Yes** | **80.72** | **67.42** | **50.65** | **43.81** | **71.13** | **58.13** | **37.02** | **50.62** |

## 4 EXPERIMENTATION

We evaluate all proposed spiking video-language models on moment retrieval and highlight detection tasks using the QVHighlights and Charades-STA datasets. Since, to the best of our knowledge, our proposed model is the first spiking VLM evaluated on VTG tasks, we benchmark its performance against state-of-the-art non-spiking video-language models. Additionally, we perform a study comparing our three model variants—Vanilla SpikingVTG, NF-SpikingVTG, and 1-bit NF-SpikingVTG— on task specific performance and computational efficiency. Preliminary energy analysis further highlights the potential benefits of each model version.

### 4.1 EXPERIMENTAL DETAILS

The Spiking Transformer core in our model comprises four encoder layers, each with a hidden dimension of 1024, with 8 attention heads. For the knowledge distillation phase, we employ a pretrained UniVTG model (Lin et al., 2023) that has been fine-tuned on our specific dataset. Additional

hyper-parameter and experimental details are provided in Appendix D. The experiments were run on a NVIDIA RTX A6000 GPU with 48GB memory.

**Dateset Details:** QVHighlights (Lei et al., 2021) is the only public dataset that includes ground-truth annotations for moment retrieval and highlight detection, allowing for a thorough evaluation of the performance of our model and additional ablation studies. We also employ the Charades-STA dataset (Gao et al., 2017) to conduct further assessments on additional moment retrieval tasks. Additional details on datasets are available at Appendix C.

**Evaluation Metrics:** For QVHighlights, following previous work (Lei et al., 2021) we use Recall@1 with IoU thresholds of 0.3, 0.5 and 0.7 and average mean average precision (mAP) as the evaluation metric for moment retrieval tasks. For highlight detection, we use mAP and HIT@1 (Lei et al., 2021), where a clip is considered a true positive if it receives a score of "Very Good" (Liu et al., 2022). For Charades-STA, we employ Recall@1 with IoU thresholds of 0.3, 0.5, and 0.7, along with the mean IoU (mIoU).

## 4.2 RESULTS

Our model outperforms non-spiking VTG models, including EaTR (Jang et al., 2023), 2D-TAN (Zhang et al., 2020), M-DETR (Lei et al., 2021), LLaViLo (Ma et al., 2023), UMT (Liu et al., 2022), QD-DETR (Moon et al., 2023) and non-pretrained UniVTG model (Lin et al., 2023). Additionally, it achieves competitive results compared to the current state-of-the-art pretrained UniVTG model. It is important to note that SpikeMba (Li et al., 2024) is not a fully spiking architecture; rather, one component of its multi-stage network uses an SNN. Our model establishes a baseline for future spiking VLM architectures on VTG tasks. The results are shown in Table 1 & 2.

Table 2: Performance comparison of our SpikingVTG model with SFG against other non-spiking VTG solutions on the evaluation set of the QVHighlights for **highlight detection task**.

| Method | SNN | QVHighlights | |
| --- | --- | --- | --- |
| | | mAP | HIT@1 |
| UniVTG+PT (Lin et al., 2023) | No | 41.34 | 68.77 |
| DVSE (Liu et al., 2015) | No | 18.75 | 21.79 |
| XML+ (Lei et al., 2021) | No | 35.38 | 55.06 |
| M-DETR (Lei et al., 2021) | No | 35.65 | 55.55 |
| EaTR (Jang et al., 2023) | No | 37.15 | 58.65 |
| M-DETR + PT (Lei et al., 2021) | No | 37.70 | 60.32 |
| UniVTG (Lin et al., 2023) | No | 38.83 | 61.81 |
| QD-DETR (Moon et al., 2023) | No | 39.13 | 63.03 |
| $R^2$ - Tuning (Liu et al., 2024) | No | 40.75 | 64.20 |
| UMT (Liu et al., 2022) | No | 39.85 | - |
| **SpikingVTG (Our Model)** | **Yes** | **40.74** | **68.32** |

Table 3: Performance comparison of the different SpikingVTG variants as highlighted in Fig. 4 on the evaluation set of QVHighlights dataset.

| Method | QVHighlights-MR | | | | QVHighlights-HL | | Activity |
| --- | --- | --- | --- | --- | --- | --- | --- |
| | @0.3 | @0.5 | @0.7 | mAP@avg | mAP | HIT@1 | |
| Vanilla Spiking Transformer | 78.65 | 65.10 | 47.46 | 42.56 | 40.60 | 67.42 | 0.41 |
| SpikingVTG without KD | 67.68 | 52.71 | 34.26 | 32.12 | 35.91 | 57.94 | 0.35 |
| SpikingVTG | **80.72** | **67.42** | **50.65** | **43.81** | **40.74** | **68.32** | 0.34 |
| NF-SpikingVTG | 79.87 | 66.45 | 48.27 | 42.68 | 40.54 | 67.61 | 0.25 |
| 1-bit NF-SpikingVTG | 78.77 | 65.16 | 47.35 | 42.32 | 40.31 | 67.29 | **0.19** |
| 1-bit NF-SpikingVTG w/ ReLU | 78.39 | 66.06 | 47.10 | 41.78 | 40.22 | 67.10 | **0.19** |

## 4.3 ABLATION STUDY

As demonstrated in Table 3, the inclusion of the Spike Feedback Gating (SFG) mechanism enhances performance compared to the model without SFG, i.e. a vanilla spiking transformer. Furthermore, as highlighted in Fig. 3 it results in reduced neuronal activity as well. KD plays a critical role in improving the performance of the model w.r.t evaluation metrics. Moreover, the computationally efficient NF-SpikingVTG model with SFG performs competitively even when compared to other state-of-the-art (SOTA) non-spiking VLMs. Although the 1-bit NF-SpikingVTG variant shows a slight reduction in performance across evaluation metrics, it is highly memory efficient and involves simpler computations, making it well-suited for deployment on resource-constrained hardware. Furthermore, Table 3 also presents the average model-wide neural activity of the spiking model, calculated over $T = 10$ time steps. This metric represents the proportion of active neurons per timestep, averaged across all layers. This demonstrates that the optimizations aimed at enhancing computational efficiency (i.e. reducing non-local normalization operation and introducing quantized weights) also effectively

reduce overall neural activity in the model. We also implement a variant of 1-bit NF SpikingVTG by replacing all GELU layers with hardware friendly ReLU layer Timcheck et al. (2023).

### 4.4 Analysis of Energy and Power Efficiency

We conduct a preliminary energy analysis of the proposed SpikingVTG variants during test-time inference and compare it to a non-spiking UniVTG model with comparable depth and hidden state dimensions (also hidden dimension ($D$) is same as intermediate layer dimension in our implementation). For this energy analysis, we focus solely on the cost of arithmetic operations, excluding the cost of memory I/O transactions. From a simpler circuit design standpoint, for our analysis we consider 45nm CMOS technology and 32-bit precision, thus floating point (fp)-MAC operations consume $4.6pJ$, fp-ACC operations consume $0.9pJ$ and integer(int)-ACC operations consume $0.1pJ$ (Han et al., 2015). The primary energy consumption is attributed to the transformer encoder layers, which consist primarily of the attention mechanism and multiple linear layers (Wang et al., 2023). The primary computation cost, calculated for an input sequence of length $L$, of each transformer encoder-layer of the non-spiking model can be expressed as: $E_A = [(3LD^2) + (LD^2 + L^2D) + (LD^2) + (LD^2)]$ fp-MAC operations, corresponding to the three projection layers, the attention mechanism, the intermediate layer, and the output layer.

For the SpikingVTG model, per spiking transformer encoder layer the computational cost per time step is given by: $E_{S_t} = [(3 \cdot IFR_{in} \cdot LD^2) + (IFR_k \cdot LD^2 + IFR_v \cdot L^2D) + (IFR_{attn} \cdot LD^2) + (IFR_{interm.} \cdot LD^2)]$ fp-ACC operations, where each term is associated for each component similar to the one specified above. $IFR_l$ represents the mean firing rate of the corresponding layer $l$. The total energy cost for the spiking model is: $E_S = (E_{S_t} * T)$ fp-ACC operations, where $T$ represents the number of time steps the model is operated. The models that include normalization also have an added cost of energy for normalization however, it is of the order $O(LD)$ so it has not been included in our computation. It is to be noted that both our NF-SpikingVTG and 1-BIT NF-SpikingVTG models are normalization free so they do not incur this added cost. Moreover, for 1-bit SpikingVTG, the core computations in matrix multiplications shift from using fp-ACC to int-ACC operations.

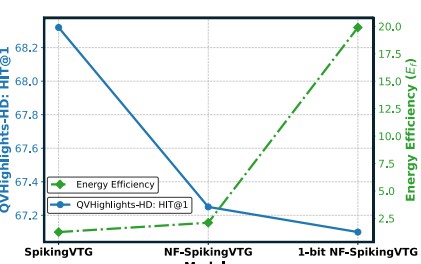

Figure 5: Graph depicting the performance of each SpikingVTG variant on the QVHighlights highlight detection task, alongside their potential energy efficiency ($E_f$).

We define energy efficiency of the spiking model as $E_f = E_A/E_S$. Specific examples illustrating energy efficiency of SpikingVTG models is provided in Appendix D. When operating the underlying models for $T = 10$ time steps, the energy efficiency and performance of each model are illustrated in Fig. 5. The average power efficiency for each model type is calculated as $P_f = \frac{(E_A/1)}{(E_S/T)} = E_f \times T$, demonstrating that our models are significantly more power-efficient (ranging from $12.5\times$ in SpikingVTG to up to $200\times$ in 1-bit NF-SpikingVTG) compared to non-spiking models. This efficiency arises from the ability of SNNs to unroll complex operations over time, thus providing low-powered solutions for complex tasks unlike conventional non-spiking architecture. Although this method of analysis does not account for architectural energy advantages, it provides a useful approximation to gauge the potential benefits of spiking models over their non-spiking counterparts.

## 5 Conclusions

Our saliency feedback gating-enabled SpikingVTG model offers a computationally efficient approach for VTG tasks while maintaining competitive performance with state-of-the-art non-spiking models. By harnessing layer-wise convergence dynamics, we efficiently train our model using implicit differentiation at equilibrium. We employ a multi-stage training pipeline that incorporates knowledge distillation, using the non-spiking pretrained UniVTG model as the "teacher" and the SpikingVTG model as the "student". This training pipeline further enables architectural optimizations, leading to the development of Normalization Free (NF)-SpikingVTG and 1-bit NF-SpikingVTG, enhancing computational efficiency and facilitating the on-chip deployment of these complex models.

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

## A  EXTENDED ARCHITECTURE OVERVIEW

### A.1  SPIKING TRANSFORMER: LAYER-WISE DETAILS

The Spiking Transformer layer primarily consists of a spiking multi-head attention (MHA) block, followed by a spiking feedforward network comprising an intermediate layer and an output layer with both inter- and intra-layer communication happening using spikes. Details of the operations in each layer are provided below.

**Spiking Attention Block:** In Spiking MHA, to enable computationally efficient accumulate based operations the input to the attention layer are spikes instead of real-valued data. The spiking attention mechanism is given as follows,

$$Attn(X_s[t], K_s[t], V_s[t]) = \phi(d * Q(X_s[t]) \cdot (K_s[t])^T) \cdot V_s(t) \tag{10}$$

Here, $Q(X_s(t))$ represents the Query, obtained by passing the input spikes $X_s(t)$ at time $t$ through a linear layer ($W_Q$). The spikes for the Key layer ($K_s(t)$) are generated by passing $X_s(t)$ through a linear mapping ($W_K$), followed by an LIF neuron layer. Similarly, we generate spikes for Value. $d$ is a scaling constant. Since the input, key, and value matrices consist of spike trains rather than real-valued data, the primary computations in all matrix multiplications are floating-point accumulation operations rather than floating point multiplicative and accumulative operations. In the NF-SpikingVTG variant, as discussed in the paper, we use $\phi$ as the $ReLU$ function, significantly reducing the computational overhead compared to employing $\phi$ as the non-local $Softmax$ operation. The output of the attention layer is fed to an LIF neuron, which outputs spikes. The convergence dynamics of the layer at equilibrium is given as, $a^*_{attn} = \sigma(\frac{1}{V_{th}}(Attn(a^*_x, a^*_k, a^*_v) + b_{attn})$, where $a_x$ represents the ASR of the layer used to generate the Query, $a_k$ denotes the ASR of the Key, and $a^*_v$ corresponds to the ASR of the Value. $b_{attn}$ is a bias term.

**Intermediate Layer:** The intermediate layer takes as input the spikes generated from the preceding layer and maps it to an intermediate dimension with a linear layer. The output is then passed through an LIF layer. The convergence dynamics of the layer at equilibrium is given as, $a^*_{interm.} = \sigma(\frac{1}{V_{th}}(gelu(W_{interm}.a^*_p) + b_{interm.}))$, where $W_{interm.}$ is the linear weight and gelu() is the activation used for the layer. $a^*_p$ is the ASR at equilibrium for the previous layer. $b_{interm.}$ is a bias term. During inference, all matrix multiplications involve accumulative operations due to the nature of the input.

**Output Layer:** The output layer takes as input the spikes generated from the preceding layers as shown in Fig. 1. The output is then passed through an LIF layer. The convergence dynamics of the layer at equilibrium is given as, $a^*_{output} = \sigma(\frac{1}{V_{th}}(norm(W_{output}a^*_{interm.} + a^*_p) + b_{output}))$, where $W_{output}$ is the linear weight and layer norm is used for normalization. $a^*_{interm.}$ is the ASR at equilibrium for the previous intermediate layer. $b_{output}$ is a bias term. During inference, all matrix multiplications involve accumulative operations due to the nature of the input. In the NF-SpikingVTG model we further remove the layer normalization to improve on-chip deployability.

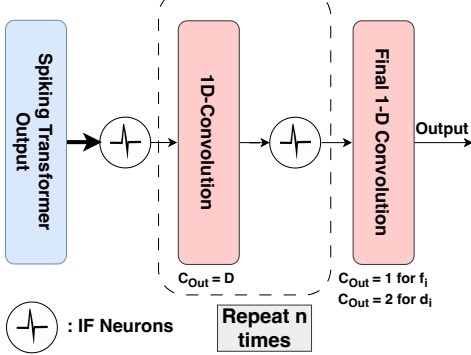

Figure 6: High-level overview of the spiking decoder.

### A.2  SPIKING DECODER

As discussed in the main paper, the spiking decoder processes the output spikes from the spiking transformer core by applying a series of 1D convolutions followed by Integrate-and-Fire (IF) neurons. This setup predicts two parameters: the foreground indicator ($f_i$) for each clip, and $d_i = [d_{s_i}, d_{e_i}]$. For $f_i$, the final layer has a single output channel, and its temporal mean is passed through a sigmoid activation to generate the prediction. For $d_i$, the final convolutional layer outputs two channels, as illustrated in Fig. 6.

## B LOSS FUNCTION DETAILS

As described in the main paper, the total loss over $N$ clips in the training set is defined as $L = \frac{1}{N} \sum_{i=1}^{N} \left( L_{f_i} + L_{d_i} + L_{c_i} \right)$, where $L_f$ represents the binary cross-entropy loss for the indicator variable $f_i$, $L_d$ combines the smooth L1 loss with the generalized IoU loss Rezatofighi et al. (2019) for the predicted boundaries, and $L_c$ is an optional loss term incorporating intra- and inter-video contrastive learning Chen et al. (2020). We follow similar loss function construction as previous works on VTG Lei et al. (2021); Lin et al. (2023). The loss for fore-ground parameter is given as follows,

$$L_f = -\lambda_f \left[ f_i \log \tilde{f}_i + (1 - f_i) \log(1 - \tilde{f}_i) \right] \tag{11}$$

where, $f_i$ is the true label and $\tilde{f}_i$ is the model prediction. The loss for predicted boundaries is given as follows,

$$L_d = \mathbf{1}_{f_i=1} \left( \lambda_{\text{L1}} L_{\text{SmoothL1}}(\tilde{d}_i, d_i) + \lambda_{\text{iou}} L_{\text{iou}}(\tilde{b}_i, b_i) \right) \tag{12}$$

where, $d_i, b_i$ are the true label and $\tilde{d}_i, \tilde{b}_i$ is the model prediction. $L_c = \lambda_{\text{inter}} L_{\text{inter}} + \lambda_{\text{intra}} L_{\text{intra}}$ is used for inter-video and intra video contranstive learning (Lin et al., 2023). For each video $V$, we randomly select a clip $v_i$ with fore-ground indicator = 1 and positive saliency score. Clips from the same video, denoted as $v_j$, with saliency scores $s_j < s_i$ are treated as negative samples. i.e., $A = \{j \mid s_j < s_i, 1 \le j \le L_v\}$, and perform intra-video contrastive learning using the loss

$$L_{\text{intra}} = -\log \frac{\exp(\tilde{s}_i/\tau)}{\exp(\tilde{s}_i/\tau) + \sum_{j \in A} \exp(\tilde{s}_j/\tau)} \tag{13}$$

. Furthermore, we treat textual queries from other samples within the batch ($k \in S$) as negative samples, enabling inter-video contrastive learning for cross-sample supervision:

$$L_{\text{inter}} = -\log \frac{\exp(\tilde{s}_i/\tau)}{\sum_{k \in S} \exp(\tilde{s}_i^k/\tau)} \tag{14}$$

, where $S$ is the training batch, $\tilde{s}_i^k = \cos(v_i, M_k)$ and $M_k$ is the sentence representation (Eqn. 4) and $cos$ is cosine similarity.

## C DATASET DETAILS

**QVHighlights:** The QVHighlights dataset Lei et al. (2021) stands out as the sole dataset providing annotations for both moment retrieval and highlight detection, making it an excellent resource for benchmarking on both the VTG tasks. Comprising 10,148 videos with an average duration of 150 seconds. It features a total of 10,310 queries linked to 18,367 moments, resulting in an average of 1.8 distinct moments per query within each video. The dataset spans a variety of scenarios, including daily vlogs, travel vlogs, and news events.

## D ADDITIONAL EXPERIMENTAL DETAILS

In this subsection, we provide a concise overview of the implementation details and provide additional experimental details. The GPU specifications for the experiments are detailed in the main paper, while the CPU utilized is an AMD Ryzen Threadripper 3960X 24-Core Processor. We have used Python and the PyTorch framework to write the code. The video and textual feature are developed following previous work (Lei et al., 2021; Lin et al., 2023). We have used the Adam optimizer to train our model. We list the hyper-parameters used in the work in Table 4. We perform 20 epochs of KD with operating time steps T = 50. The memory requirement is 25GB considering batch size of 32 and QVHighlights dataset. The clock time for 20 epochs of KD on 1 NVIDIA RTX A6000 GPU with 48GB memory was around 2 hours.

| Hyper-parameters | Range | Optimal |
|---|---|---|
| $N$: Encoder Layers | (2-6) | 4 |
| $D$: Hidden Dimension | (768-2048) | 1024 |
| $n_1$: $f$-decoder depth | (1-5) | 3 |
| $k_1$: $f$-decoder kernel size | (3-9) | 3 |
| $n_2$: $d$-decoder depth | (1-5) | 3 |
| $k_2$: $d$-decoder kernel size | (3-9) | 7 |
| $T_{KD}$: Timesteps for KD | (5-100) | 50 |
| $T_f$: Timesteps for Finetuning | (5-50) | 10 |
| $V_{th}$: Threshold Potential | (0.5 - 2.0) | 1.0 |
| $\gamma$: Leaky-factor | (0.9 - 1.0) | 0.99 |
| $\lambda_f$ :$L_f$ co-efficient | (1 - 20) | 10 |
| $\lambda_{L1}$ :$L_{SmoothL1}$ co-efficient | (1 - 20) | 10 |
| $\lambda_{intra}$ :$L_{intra}$ co-efficient | (0 - 1.0) | 0.05 |
| $\lambda_{inter}$ :$L_{inter}$-co-efficient | (0 - 1.0) | 0.01 |
| $\lambda_{iou}$ :$L_{iou}$ co-efficient | (1 - 20) | 10 |
| $lr$: Learning Rate | $(1e^{-5} - 1e^{-6})$ | $8e^{-6}$ |
| $w_d$: weight decay | $(1e^{-5} - 1e^{-3})$ | $1e^{-4}$ |
| Batch Size | (8-64) | 32 |
| Epochs: KD | 10-50 | 20 |
| Epochs: Finetuning | 20-200 | 100 |

Table 4: Hyper-parameters of our SpikingVTG model w/SFG. Optimal values for QVHighlights dataset is also shown.

**Charades-STA:** The Charades-STA dataset comprises 16,128 indoor videos, each with an average duration of 30.6 seconds. It includes 12,408 query-interval pairs designated for training and 3,720 query-interval pairs reserved for testing.

### D.1 VISUALIZING SFG MECHANISM

As highlighted in Section. 3.2.3, the SFG mechanism computes dynamic saliency score ($F_s^{v_i}[t]$), at time t, for the i-th segment of the video. As shown in Fig. 7, we analyze the scores at equilibrium to gain insights into the functioning of the SFG enabled multiplicative gating mechanism. The clips neighboring the clip of interest (for highlight detection task) show higher scores at equilibrium, highlighting the effectiveness of the SFG mechanism.

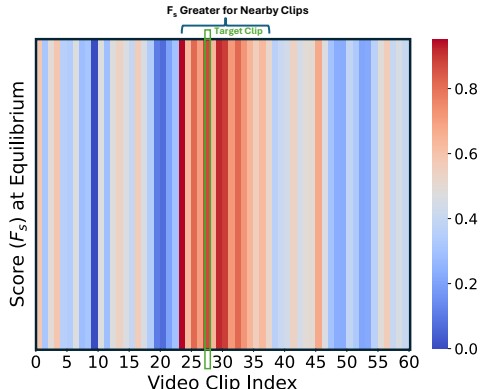

Figure 7: Heatmap showing the scores per clip ($F_s$) at equilibrium, with the target frame for highlight detection corresponding to clip index 28.

### D.2 COMPARING IMPLICIT DIFFERENTIATION AT EQUILIBRIUM WITH BPTT

We were unable to train our model using BPTT due to its significantly higher memory demands during training. For example, training our SpikingVTG model on the QVHighlights dataset requires 25GB of memory with a batch size of 32 and 50 time steps for convergence (T). In contrast, the memory requirement for training with BPTT exceeds 100GB, as it requires storing the entire computational graph throughout the training process. Consequently, training the model with BPTT is not feasible. This also underscores the advantage of the equilibrium-based training mechanism employed in this work.

### D.3 ADDITIONAL EXPERIMENTS

We perform additional experiments on TaCOS dataset Regneri et al. (2013) for moment retrieval tasks and Youtube Highlights dataset Sun et al. (2014) for highlight detection tasks. We present the results in Table 5 and 6.

Table 5: Performance comparison of our SpikingVTG models against non-spiking VTG solutions on the evaluation set of TaCOS dataset.

| Method | TaCOS | | | |
|---|---|---|---|---|
| | @0.3 | @0.5 | @0.7 | mIoU |
| UniVTG+PT | 56.11 | 43.44 | 24.27 | 38.63 |
| 2D TAN | 40.01 | 27.99 | 12.92 | 27.22 |
| VSLNet | 35.54 | 23.54 | 13.15 | 24.99 |
| MDETR | 37.97 | 24.67 | 11.97 | 25.49 |
| UniVTG | 51.44 | 34.97 | 17.35 | 33.60 |
| SpikingVTG w/o SFG | 52.83 | 37.39 | 20.03 | 34.17 |
| **SpikingVTG** | 54.32 | 39.16 | 21.78 | 35.78 |

Table 6: Performance comparison of our SpikingVTG model with SFG against non-spiking VTG solutions on the evaluation set of Youtube Highlights dataset.

| Method | Youtube-HL | | | | |
|---|---|---|---|---|---|
| | Dog | Gym. | Skating | Skiing | Avg |
| UniVTG+PT | 74.3 | 79.0 | 84.9 | 75.1 | 78.6 |
| QD-DETR | 72.2 | 77.4 | 72.7 | 72.8 | 74.4 |
| UniVTG | 71.8 | 76.5 | 73.3 | 73.2 | 75.2 |
| MINI-Net | 58.2 | 61.7 | 72.2 | 58.7 | 64.4 |
| Joint-VA | 64.5 | 71.9 | 62.0 | 73.2 | 71.8 |
| UMT | 65.9 | 75.2 | 71.8 | 72.3 | 74.9 |
| SpikeMba | 74.4 | 75.4 | 74.3 | 75.5 | 75.5 |
| **SpikingVTG** | 73.9 | 78.1 | 80.1 | 74.2 | 76.6 |

## D.4 ENERGY ANALYSIS

Let us walk through a specific example of analyzing the energy consumption for the 1-bit NF-SpikingVTG model. Consider a single transformer encoder layer. As discussed in the main paper, the computational cost of the layer for a non-spiking model is given by: $E_A = [3LD^2 + (LD^2 + L^2D) + LD^2 + LD^2] * (4.6mJ)$. In our implementation, we use $D = 1024$ and lets consider total sequence length $L = 200$. Thus we get energy cost of each block is $5.98mJ$. Now, energy cost per time step of our 1-bit NF-SpikingVTG is given as, $E_{S_t} = [(3 \cdot IFR_{in} \cdot LD^2) + (IFR_k \cdot LD^2 + IFR_v \cdot L^2D) + (IFR_{attn} \cdot LD^2) + (IFR_{interm.} \cdot LD^2)] * (0.1pJ)$. Empirically we find $IFR_{in} = 0.40, IFR_k = 0.18, IFR_v = 0.19, IFR_{attn} = 0.03, IFR_{interm.} = 0.09$.

Thus $E_{S_t} = 0.03mJ$ resulting in $E_S = E_{S_t} * T = .3mJ$, where T = 10. Thus efficiency factor is $E_f = E_A/E_S = 19.93$.

