# OpenReview forum: "SpikingVTG: Saliency Feedback Gating Enabled Spiking Video Temporal Grounding"
_ICLR.cc/2025/Conference — Submitted to ICLR 2025_

### Official Review · Reviewer_zh6n · 2024-10-17

**Soundness:** 3
**Presentation:** 2
**Contribution:** 2
**Rating:** 6
**Confidence:** 4

**Summary:**

This paper presents SPIKINGVTG, a SNN based Multimodel Transformer model for Video Temporal Grounding tasks. The author proposes a series of works to improve energy efficiency while reducing performance loss.

Specifically, the core contribution includes:

1.Architecturally: SALIENCY FEEDBACK GATING (SFG): using intermediate temporal outputs to dynamically update the input to the model at every time step for better predictions.

2.Theoretical analysis: TRAINING LEVERAGING CONVERGENCE DYNAMICS: the author shows that the layer-wise ASRs converge to equilibrium, enabling the derivation of steady-state equations for linear layers

3.Training: MULTI-STAGED TRAINING PIPELINE: A series of work to simplify the model and improve its energy efficiency
-step1: KNOWLEDGE DISTILLATION from Non-spiking model (UniVTG)
-step2: supervised Finetune on the tasks
-step3: REPLACING SOFTMAX AND REMOVING LAYER NORMALIZATION
-step4: 1-BIT WEIGHT QUANTIZED SPIKINGVTG

The proposed method achieves comparable performance on QVHighlights and Charades-STA.

**Strengths:**

There're a few strenghts:
1. As far as I know, this is the first paper using SNN for VTG tasks.
2. The author proposed SALIENCY FEEDBACK GATING to preprocess multi-modal information during input. The specific method is to use the features output by the model to calculate Cosine Similarity for the Text Features features, and weight the Video Features based on the results. According to the authors, this module improves performance and reduces overall neural activity.
3. During training, the author uses CONVERGENCE DYNAMICS at equilibrium to reduce BPTT's dependence on the intermediate calculation graph.
4. In terms of model efficiency, the author fully integrates existing methods, including model distillation, softmax removal, layernorm removal, 1-bit model quantization, etc. Experimental results show that these operations do not lead to significant performance degradation, but fully improve the model's energy efficiency and deployability on neuromorphic chips.

Overall, I think this paper reveals the possibility of using SNN for low-energy VTG.

**Weaknesses:**

A. Methodology
Although the authors integrated many methods and first revealed the feasibility of VTG with SNN, I find it difficult to see a clear research question and clear original contribution besides SALIENCY FEEDBACK GATING (less than one page). Specifically:

1.I think SFG is an interesting and original module that uses the model’s output as a gating masked video input to enhance the model’s attention to key video segments. But the author lacks more demonstration of its effectiveness, motivation and scalability, and the main explanation is less than one page. Some questions are as follows:
-Can the author perform a visualization similar to gradcam to show the effectiveness of gating? (https://arxiv.org/abs/1610.02391)
-Does feedback connections participate in gradient backpropagation?
-The symbols and meaning of Eq4 need to be more clearly explained. I can read it, but the meaning of many symbols is not clearly pointed out.
-When calculating the dot product, how to ensure that the text feature M encoded by a single layer network and the video feature a encoded by a deep network are in the same feature space and can produce meaningful similarity measurements?
-Does this part of the module also perform subsequent 1-bit quantization? That is, weight values ​​in {−1, 1} and activations in ∈ {−1, 0, 1}?


2.Lack of novelty:
Except for SFG, most of the methods in this article come from previous work, including: spiking transformer, ternary spiking model,  knowledge distillation, removing softmax and layer norm, and 1-bit quantization.

Some research backgrounds are not well introduced, such as Softmax free (ANN/SNN) transformer, eg:
https://arxiv.org/html/2206.08898v2
https://arxiv.org/pdf/2409.15375

These integrations make sense, but I struggle to see original contributions beyond demonstrating SNN feasibility.

3.Lack of Experiment
The author only conducted experiments on two datasets, QVHighlights and Charades-STA. This is a small part of the baseline UniVTG. Can the author add more baselines for comparison? See
https://arxiv.org/abs/2307.16715

**Questions:**

See weakness:
-Can the author perform a visualization similar to gradcam to show the effectiveness of gating? (https://arxiv.org/abs/1610.02391)

-Does feedback connections participate in gradient backpropagation?

-The symbols and meaning of Eq4 need to be more clearly explained. I can read it, but the meaning of many symbols is not clearly pointed out.

-When calculating the dot product, how to ensure that the text feature M encoded by a single layer network and the video feature a encoded by a deep network are in the same feature space and can produce meaningful similarity measurements?

-Does this part of the module also perform subsequent 1-bit quantization? That is, weight values ​​in {−1, 1} and activations in ∈ {−1, 0, 1}?

-Can the author add more baselines for comparison?

More Questions:
I am quite impressed by the accuracy of the 1-bit model shown by the author in table 3. Combined with SNN, quantization technology is expected to significantly reduce the size, calculation amount and energy consumption of the model. But I'm curious why the model's performance doesn't drop significantly at 1bit without doing additional operations? The author seems to be quite close to the ann full-precision baseline of UniVTG+PT when using W in {−1, 1} and activations in {−1, 0, 1}? Does SNN promote 1-bit quantization here? More specifically:
-If the author trains a 1-bit ann of the same size using the same method and training pipeline, how does the performance compare with SNN?
-For table3, can a more comprehensive ablation experiment be performed on 1-bit to reveal more information? Is the high performance just because Bitnet's (https://arxiv.org/pdf/2310.11453) is good enough, or is there something in the paper that brings additional benefits?
-In addition, please include the important information of the size of the model in the paper and results, especially the comparison before and after ann-snn quantization.

---

> ### Author Response · Authors · 2024-11-22
>
> Thank you for your detailed and valuable feedback on our paper. In this rebuttal, we address your comments below.
>
>
> **Comment 1:**
> >1. Methodology Although the authors integrated many methods and first revealed the feasibility of VTG with SNN, I find it difficult to see a clear research question and clear original contribution besides SALIENCY FEEDBACK GATING (less than one page). Specifically,
>
> >**Comment 1.a:** Can the author perform a visualization similar to gradcam to show the effectiveness of gating?
>
> **Response:**
> Thank you for this excellent suggestion. In the revised paper, we have included a section in the Appendix (Sec. D.1) to visualize the scores $F_s$ (Eqn. 4)  assigned by the SFG to each video clip. This visualization clearly highlights how clips in the vicinity of the target clip or interval are given higher importance. We believe this addition strengthens our contributions and provides deeper insights into the proposed approach.
>
>
> > **Comment 1.b**
> Does feedback connections participate in gradient backpropagation?
> The symbols and meaning of Eq4 need to be more clearly explained. I can read it, but the meaning of many symbols is not clearly pointed out.
>
> **Response:**
> Yes, the feedback connections also contribute to gradient backpropagation.
> We have revised the discussion around Equation 4 to further clarify it. Thank you for bringing it to our attention.
>
> > **Comment 1.c:**
> When calculating the dot product, how to ensure that the text feature M encoded by a single layer network and the video feature encoded by a deep network are in the same feature space and can produce meaningful similarity measurements?
>
> **Response:**
> Thank you for this question. We first compute $M$, which is a representation of the sentence and represent it in the same dimension as individual clips in the video. We compute $\mathbf{M} = \mathbf{Q}^TSoftmax(\mathbf{Q}\mathbf{W_p})$ where, $\mathbf{M} \in \mathbb{R}^{D}$, textual query features $\mathbf{Q} \in \mathbb{R}^{L_q \times D}$, input video features $V \in \mathbb{R}^{L_v \times D}$ and  $\mathbf{W_p} \in \mathbb{R}^{D \times 1}$ is a learnable embedding. Now, since $\mathbf{M}$ and individual clips of the video are in the same dimension we can perform the operation as highlighted in Eqn. 4. We have clarified this section in the revised manuscript for better understanding.
>
>  >**Comment 1.d:** Does this part of the module also perform subsequent 1-bit quantization? That is, weight values ​​in {−1, 1} and activations in ∈ {−1, 0, 1}?
>
> **Response:**
> Yes, we apply 1-bit quantization to $W_p$​, utilized in the SFG mechanism. Additionally, the outputs of the SFG mechanism are processed through an LIF layer to generate spikes, as illustrated in Fig. 1. Furthermore, for the Norm-Free variant (NF-SpikingVTG), we replace the Softmax function (for computing $M$) with the ReLU-based alternative outlined in Section 3.4.2.

---

> ### Author Response · Authors · 2024-11-22
>
> **Comment 2:**
> >2.Lack of novelty: Except for SFG, most of the methods in this article come from previous work, including: spiking transformer, ternary spiking model, knowledge distillation, removing softmax and layer norm, and 1-bit quantization.
>
> **Response:**
> Thank you for this comment. We would like to emphasize the primary contribution of this work.
>
> (a) We propose the Saliency Feedback Gating (SFG) mechanism which leverages the temporal dynamics of SNN architectures to perform a multiplicative gating mechanism on the sequence of clips in the video, resulting not only in improved performance over a vanilla spiking transformer but also considerably improving computational efficiency by reducing model-wide neuronal activity (4.3 Ablation Study). Visualization of SFG mechnaism is also added in Appendix D.1.
>
> (b) We demonstrate that incorporating the SFG mechanism preserves the model's convergence dynamics (Figure 3). This stable convergence to equilibrium enables us to leverage implicit differentiation at equilibrium as a robust training framework. Moreover, it facilitates the knowledge distillation from the intermediate states of a 'teacher' ANN to the converged intermediate states of a 'student' spiking model, as discussed in Section 3.4.1. While implicit differentiation at equilibrium has been previously explored, as noted by the reviewer, its application was largely restricted to simpler datasets like CIFAR-10/100. For the first time, we scale this approach to multimodal transformer-based SNNs, empowered by the SFG gating mechanism, and empirically demonstrate the convergence dynamics of the underlying model (Figure 3).
>
> (c) This paper introduces, for the first time, a practically viable multi-modal spiking neural network (SNN) architecture that eliminates non-local normalization operations and can undergo extreme weight quantization. Non-local operations like layer normalization and softmax hinder the deployment of many spiking architectures on neuromorphic chips. As detailed in Section 3.4, these model optimizations are achievable only through our multi-staged training pipeline, which integrates an equilibrium-based knowledge distillation (KD) mechanism. Below, we present results comparing a 1-bit, norm-free SpikingVTG model trained from scratch with a variant trained using our proposed multi-staged pipeline. These comparisons highlight the non-trivial nature of the training process introduced in our work.
>
> | Model Type | R1@0.3 | R1@0.5 | R1@0.7 | mAP@avg | mAP | HIT@1 |
> |-------------|--------|--------|--------|---------|------|-------|
> | 1-bit NF-SpikingVTG from Scratch | 66.71 | 50.16 | 32.17 | 31.34 | 34.13 | 56.21 |
> | 1-bit NF-SpikingVTG Multi-Staged pipeline | 78.77 | 65.16 | 47.35 | 42.32 | 40.31 | 67.29 |
>
>
>  **Comment 3:**
> >Some research backgrounds are not well introduced, such as Softmax free (ANN/SNN) transformer, eg: https://arxiv.org/html/2206.08898v2 https://arxiv.org/pdf/2409.15375
>
> **Response:**
> Thank you for highlighting these works. We have revised our paper and discussed them in the Introduction section.
> While the aforementioned works enable softmax-free transformers, it is important to note that they still rely on non-local layer normalization operations. Moreover, their focus is primarily on vision-based data, unlike our framework, which addresses multi-modal video and language data.

---

> ### Author Response · Authors · 2024-11-22
>
> **Comment 4:**
> >3.Lack of Experiment The author only conducted experiments on two datasets, QVHighlights and Charades-STA. This is a small part of the baseline UniVTG. Can the author add more baselines for comparison? See https://arxiv.org/abs/2307.16715
>
>
> **Response:**
> Thank you for this comment. Following your suggestion we have also evaluated our model on two more dataset, viz TACoS for moment retrieval and Youtube HL for highlight detection. We have updated the paper and also added the results below for your reference.
>
>
> Results on TaCOS:
>
>
> | Model Type     | R1@0.3 | R1@0.5 | R1@0.7 | mIoU  |
> |----------------|--------|--------|--------|-------|
> | 2D TAN         | 40.01  | 27.99  | 12.92  | 27.22 |
> | VSLNet         | 35.54  | 23.54  | 13.15  | 24.99 |
> | MDETR          | 37.97  | 24.67  | 11.97  | 25.49 |
> | UniVTG         | 51.44  | 34.97  | 17.35  | 33.60 |
> |EMTM | 45.78 | 34.83 | 23.41 | 34.44 |
> | SpikingVTG     | 54.32  | 39.16  | 21.78  | 35.78 |
>
>
> Results on YoutubeHL (mAP):
>
>
> | Model Type   | Dog   | Gym.  | Skating | Skiing | Avg   |
> |--------------|-------|-------|---------|--------|-------|
> | QD-DETR      | 72.2  | 77.4  | 72.7    | 72.8   | 74.4  |
> | UniVTG       | 71.8  | 76.5  | 73.3    | 73.2   | 75.2  |
> | MINI-Net     | 58.2  | 61.7  | 72.2    | 58.7   | 64.4  |
> | Joint-VA     | 64.5  | 71.9  | 62.0    | 73.2   | 71.8  |
> | UMT          | 65.9  | 75.2  | 71.8    | 72.3   | 74.9  |
> | SpikeMba     | 74.4  | 75.4  | 74.3    | 75.5   | 75.5  |
> | SpikingVTG   | 73.90 | 78.07 | 80.10   | 74.20  | 76.55 |
>
>  **Comment 5:**
> >Can the author add more baselines for comparison?
>
> **Response:**
> Thank you for this suggestion. We have addressed this point in our response to Comment 4.
>
>  **Comment 6:**
> >More Questions: I am quite impressed by the accuracy of the 1-bit model shown by the author in table 3. Combined with SNN, quantization technology is expected to significantly reduce the size, calculation amount and energy consumption of the model. But I'm curious why the model's performance doesn't drop significantly at 1bit without doing additional operations? The author seems to be quite close to the ann full-precision baseline of UniVTG+PT when using W in {−1, 1} and activations in {−1, 0, 1}? Does SNN promote 1-bit quantization here? More specifically: -If the author trains a 1-bit ann of the same size using the same method and training pipeline, how does the performance compare with SNN? -For table3, can a more comprehensive ablation experiment be performed on 1-bit to reveal more information? Is the high performance just because Bitnet's (https://arxiv.org/pdf/2310.11453) is good enough, or is there something in the paper that brings additional benefits? -In addition, please include the important information of the size of the model in the paper and results, especially the comparison before and after ann-snn quantization.
>
> **Response:**
>
> Thank you for highlighting this point. Our 1-bit NF-SpikingVTG variant, not only incorporates 1-bit weights but also do not use any non-local operations (layer norm. / softmax). BitNet explored quantizing a full-precision ANN to 1-bit weights while retaining all normalization operations. However, when we attempted to remove layer normalization from the underlying ANN while introducing 1-bit weights, the ANN's performance deteriorated significantly. Furthermore, the 1-bit NF-SpikingVTG model not only achieves performance comparable to the full-precision model but also exhibits significantly reduced neural activity (as highlighted in Section 4.3, Ablation Study). This advantage is uniquely realized in an SNN setting, as shown in Figure 5. Our full-precision model requires 322 MB of memory, whereas the 1-bit variant, after converting all 32-bit linear layers to 1-bit layers, consumes approximately 11 MB.
>
> **Thank you for your review and comments. In light of the revisions that we have now made in response to your comments, we kindly request that you reconsider your rating. We are happy to address any further feedback you may have.**

---

> > ### Comment · Reviewer_zh6n · 2024-11-22
> > **Rating increased from 5 to 6**
> >
> > Thanks to the author for the reply. These answers largely solved my questions. Overall I think this article is valuable, even if the essence of the advantages of SNN quantization is not yet clear. I have raised my rating from 5 to 6.

---

> > > ### Author Response · Authors · 2024-11-23
> > >
> > > We truly appreciate your decision to increase your score and are deeply grateful for the positive feedback that has helped us improve our work.

---

### Official Review · Reviewer_pBmG · 2024-11-02

**Soundness:** 2
**Presentation:** 3
**Contribution:** 2
**Rating:** 6
**Confidence:** 4

**Summary:**

This paper presents SpikingVTG, the first spiking neural network (SNN) approach to Video Temporal Grounding (VTG). The work introduces several key points: (1) a saliency feedback gating mechanism that improves performance while reducing neural activity, (2) a multi-stage training pipeline enabling efficient model variants, and (3) normalization-free and quantized versions for resource-constrained deployment. The authors demonstrate competitive performance with state-of-the-art models while achieving significant improvements in energy efficiency.

**Strengths:**

- First implementation of Video Temporal Grounding (VTG) task.
- Significant energy efficiency improvements with spiked-based operations and 1-bit precision
- Comparable performance with the previous methods
- Easy understanding with straight-forward figures and equations

**Weaknesses:**

- Only two datasets are used (QVHighlights and Charades-STA). Authors can show one more dataset such as TACoS if it is available.
- Training overhead coming from knowledge distillation (Connected to Question #3)
- Lack of results comparison with efficient VTG papers (Connected to Question #4)

**Questions:**

1. Some previous SNN papers used ternary spikes [1,2] like SpikeVTG. I just wondered if there is any result about binary spikes to show the effectiveness of ternary spikes.
2. In previous spike-based Transformer architecture, there is no need to use the Softmax function due to binarized QKV. However, this work still uses the activation function in self-attention like ReLU. Could you explain why your work should use the activation function?
3. Knowledge distillation with the ANN model should cause a huge training overhead. I wonder if the overhead, such as training time or memory, has been analyzed.
4. There are some works about efficient algorithms for VTG. It would be better to compare with [3,4].

[1] Guo, Yufei, et al. "Ternary spike: Learning ternary spikes for spiking neural networks." Proceedings of the AAAI Conference on Artificial Intelligence. Vol. 38. No. 11. 2024.
[2] Xing, Xingrun, et al. "SpikeLM: Towards General Spike-Driven Language Modeling via Elastic Bi-Spiking Mechanisms." arXiv preprint arXiv:2406.03287 (2024).
[3] Liang, Renjie, et al. "Efficient temporal sentence grounding in videos with multi-teacher knowledge distillation." arXiv preprint arXiv:2308.03725 (2023).
[4] Liu, Ye, et al. "$ R^ 2$-Tuning: Efficient Image-to-Video Transfer Learning for Video Temporal Grounding." arXiv preprint arXiv:2404.00801 (2024).

---

> ### Author Response · Authors · 2024-11-22
>
> Thank you for your detailed and valuable feedback on our paper. In this rebuttal, we address your comments below.
>
> **Comment 1:**
> >Only two datasets are used (QVHighlights and Charades-STA). Authors can show one more dataset such as TACoS if it is available.
>
> **Response:**
> Thank you for this comment. Following your suggestion we have also evaluated our model on two more dataset, viz TACoS for moment retrieval and Youtube HL for highlight detection. We have updated the paper and also added the results below for your reference.
>
> Results on TaCOS:
>
> | Model Type     | R1@0.3 | R1@0.5 | R1@0.7 | mIoU  |
> |----------------|--------|--------|--------|-------|
> | 2D TAN         | 40.01  | 27.99  | 12.92  | 27.22 |
> | VSLNet         | 35.54  | 23.54  | 13.15  | 24.99 |
> | MDETR          | 37.97  | 24.67  | 11.97  | 25.49 |
> | UniVTG         | 51.44  | 34.97  | 17.35  | 33.60 |
> |EMTM | 45.78 | 34.83 | 23.41 | 34.44 |
> | SpikingVTG     | 54.32  | 39.16  | 21.78  | 35.78 |
>
> Results on YoutubeHL (mAP):
>
> | Model Type   | Dog   | Gym.  | Skating | Skiing | Avg   |
> |--------------|-------|-------|---------|--------|-------|
> | QD-DETR      | 72.2  | 77.4  | 72.7    | 72.8   | 74.4  |
> | UniVTG       | 71.8  | 76.5  | 73.3    | 73.2   | 75.2  |
> | MINI-Net     | 58.2  | 61.7  | 72.2    | 58.7   | 64.4  |
> | Joint-VA     | 64.5  | 71.9  | 62.0    | 73.2   | 71.8  |
> | UMT          | 65.9  | 75.2  | 71.8    | 72.3   | 74.9  |
> | SpikeMba     | 74.4  | 75.4  | 74.3    | 75.5   | 75.5  |
> | SpikingVTG   | 73.90 | 78.07 | 80.10   | 74.20  | 76.55 |
>
>  **Comment 2:**
> >Training overhead coming from knowledge distillation (Connected to Question #3)
>
> **Response:**
> The knowledge distillation is only performed once at the beginning of the training as shown in Fig. 4. We perform 20 epochs of KD with operating time steps T = 50. The memory requirement is 25GB considering batch size of 32 and QVHighlights dataset. The clock time for 20 epochs of KD on 1  NVIDIA RTX A6000 GPU with 48GB memory was around 2 hours. We have added this to the Appendix D.
>
>  **Comment 3:**
> >Lack of results comparison with efficient VTG papers (Connected to Question #4)
>
> **Response:**
> Thank you for the valuable feedback. Following the reviewer's suggestion, we have compared our model with the efficient VTG works highlighted by the reviewer and updated the paper to reflect these findings. However, the SpikingVTG proposed in our work is the first spiking solution proposed for VTG so all the baseline that we have used for comparison are conventional ANN based models.
>
> **Comment 4:**
> >Some previous SNN papers used ternary spikes like SpikeVTG. I just wondered if there is any result about binary spikes to show the effectiveness of ternary spikes.
>
> **Response:**
> Thank you for highlighting this aspect. Yes, in our experiments we saw that ternary spikes achieved slightly better performance over binary spikes. Furthermore, since ternary spike does not introduce floating-point matrix multiplications in our computation we decided to use it for all our models. Also, we noticed using ternary spikes results in faster convergence during training. Below are the results evaluated on QVHighlights dataset.
>
> |Model Type |  R1@0.3 |   R1@0.5 |   R1@0.7 |  mAP@avg |  @mAP |  HIT@1 |
> |-----------------------------|---------|---------|---------|---------|-------|-------|
> |SpikingVTG with binary spikes | 80.19 | 66.97| 50.13| 43.24| 40.59 |67.56|
> |SpikingVTG with ternary spikes | 80.72 |  67.42 |  50.65 |  43.81 |  40.74 |  68.32 |
>
> **Comment 5:**
> >In previous spike-based Transformer architecture, there is no need to use the Softmax function due to binarized QKV. However, this work still uses the activation function in self-attention like ReLU. Could you explain why your work should use the activation function?
>
> **Response:**
> In our experiments we have also tried without using any activation function. There were two reasons why we decided to go with Relu. (a) Relu is implementable on a neuromorphic chips [1]  so it does not add any significant overhead during hw implementation.(b) Relu and the simple scaling factor (L^-1 as introduced in section 3.4.2), results in more stable learning.
>
> **Comment 6:**
> >Knowledge distillation with the ANN model should cause a huge training overhead. I wonder if the overhead, such as training time or memory, has been analyzed.
>
> **Response:**
> Thank you for this comment. We have addressed this point in our response to **Comment 2**.
>
> **Comment 7:**
> >There are some works about efficient algorithms for VTG. It would be better to compare with [3,4].
>
> **Response:**
> Thank you for this comment. We have addressed this point in our response to **Comment 3**.
>
> References:
>
> [1] Timcheck, Jonathan, Sumit Bam Shrestha, Daniel Ben Dayan Rubin, Adam Kupryjanow, Garrick Orchard, Lukasz Pindor, Timothy Shea, and Mike Davies. "The Intel neuromorphic DNS challenge." Neuromorphic Computing and Engineering 3, no. 3 (2023): 034005.

---

> > ### Author Response · Authors · 2024-11-22
> >
> > **Thank you for your review and comments. In light of the revisions that we have now made in response to your comments, we kindly request that you reconsider your rating. We are happy to address any further feedback you may have.**

---

> > > ### Comment · Reviewer_pBmG · 2024-11-23
> > >
> > > I appreciate your response and extra experiments. Most of the concerns have been addressed, but I still have one question.
> > >
> > > - I wonder how much performance gain is achieved when using ReLU as an activation function in self-attention, compared to non-activation operation?

---

> ### Author Response · Authors · 2024-11-23
>
> Thank you for reviewing our rebuttal. We have addressed your remaining comment below.
>
> **Comment 8:**
> >I wonder how much performance gain is achieved when using ReLU as an activation function in self-attention, compared to non-activation operation?
>
> **Response:**
> Thank you for this insightful comment. Though, using ReLU activation with scaling by $L^{−1}$ showed improvements in evaluation metrics compared to using no activation (in attention), the primary advantage was significantly faster convergence. For QVHighlights, ReLU-based activation enabled the validation loss to converge within just 25 epochs of fine-tuning, whereas the no-activation scenario required 82 epochs. It is to be noted that our model allows for **ternary spikes {-1, 0, 1}** and NF-SpikingVTG model excluded all normalization operations, including layer normalization and softmax. We will also add this discussion to the paper. Results on Highlight Detection task of QVHighlights dataset given below,
>
> |Model Type | mAP |   HIT@1 |
> |-----------------------------|---------|---------|
> |1-BIT NF-SpikingVTG ( no act in attn.)  |  39.91 | 66.54 |
> |1-BIT NF-SpikingVTG ( Relu in attn.)  |  40.31 | 67.29 |
>
> We are happy to address any further feedback you may have.

---

> > ### Comment · Reviewer_pBmG · 2024-11-24
> >
> > Thank you for your explanation. The SNN community would appreciate if the authors added this comparison in the paper for better understanding. I have decided to raise my score from 5 to 6.

---

> > > ### Author Response · Authors · 2024-11-24
> > >
> > > We truly appreciate your decision to increase your score and are deeply grateful for the positive feedback that has helped us improve our work.

---

### Official Review · Reviewer_3weo · 2024-11-04

**Soundness:** 3
**Presentation:** 2
**Contribution:** 2
**Rating:** 6
**Confidence:** 4

**Summary:**

This paper presents SpikingVTG, a new SNN architecture for Video Temporal Grounding (VTG). This paper proposes Saliency Feedback Gating (SFG) mechanism to improve the alignment of video segments with text queries, a multi-stage training pipeline that leverages knowledge distillation from a teacher model, and removing traditional neural network operations like softmax and layer normalization, thus enhancing its deployability on neuromorphic hardware.

**Strengths:**

1. Introducing a spiking neural network for video temporal grounding is original and can potentially bridge a gap in VTG with energy-efficient models. Using SNNs for temporal data aligns well with the temporal dynamics needed for VTG.

2. The inclusion of a 1-bit quantized variant enhances the practicality of deploying the model on resource-limited devices.

**Weaknesses:**

1. The presentation covers various aspects of spiking and non-spiking VTG models but lacks clarity in sections describing critical architectural details. For example, the specific spiking transformer backbone, video feature extraction method, and the text encoder used. It remains unclear whether the model performance significantly relies on an ANN for video features (e.g., a large foundational ANN model - CLIP's visual encoder).  How essential are the ANN features in terms of obtaining you final results?

2. **Limited Ablation Studies**: The paper claims significant contributions from the SFG mechanism, Normalization-Free transformer, and knowledge distillation but lacks ablation studies to isolate their effects. For instance, a comparison of the SFG mechanism against vanilla spiking transformers would better illustrate its impact on model performance and efficiency. Similarly, experiments exploring the effect of softmax and layer normalization in the spiking attention module, effect of ANN features obtained from vidual encoder, performance w/wo knowledge distillation, comparison between implicit differentiation vs. BPTT, would further support the claimed efficiency benefits.

**Questions:**

Refer to weaknesses.

---

> ### Author Response · Authors · 2024-11-15
>
> Thank you for your detailed and valuable feedback on our paper. In this rebuttal, we address your comments below.
>
>  **Comment 1:**
> > The presentation covers various aspects of spiking and non-spiking VTG models but lacks clarity in sections describing critical architectural details. For example, the specific spiking transformer backbone, video feature extraction method, and the text encoder used. It remains unclear whether the model performance significantly relies on an ANN for video features (e.g., a large foundational ANN model - CLIP's visual encoder). How essential are the ANN features in terms of obtaining your final results?
>
> **Response:**
> Thank you for this feedback. We originally included the architectural details of the spiking transformer in Appendix A (Spiking Transformer: Layerwise Details) and Table 4 (Appendix D). As you rightly pointed out, this information is critical, and to enhance accessibility, we have now moved relevant details to Section 3.2.2 (Spiking Transformer Core) in the main text while keeping additional details in Appendix A (Spiking Transformer: Layerwise Details) and Table 4 (Appendix D).
> For video feature extraction, we followed the same process as the baseline methods (Appendix D), including M-DETR, UMT, UniVTG, SpikeMba, etc. This approach ensures a fair comparison with the baselines. While we used a CLIP-based image encoder for feature extraction (similar to the compared baselines), it is worth noting that it encodes individual frames independently and does not account for the interdependencies between frames in a video sequence.
>
> > **Comment 2:**  Limited Ablation Studies: The paper claims significant contributions from the SFG mechanism, Normalization-Free transformer, and knowledge distillation but lacks ablation studies to isolate their effects. For instance, a comparison of the SFG mechanism against vanilla spiking transformers would better illustrate its impact on model performance and efficiency. Similarly, experiments exploring the effect of softmax and layer normalization in the spiking attention module, effect of ANN features obtained from visual encoder, performance w/wo knowledge distillation, comparison between implicit differentiation vs. BPTT, would further support the claimed efficiency benefits.
>
> **Response:** Thank you for emphasizing the importance of ablation studies. In the revised paper, we have included a comprehensive ablation study and outlined the key findings below for clarity.
>
>  **A. Effect of SFG mechanism** Building on the reviewer’s insightful observation, we have compared our SpikingVTG model with the SFG mechanism enabled against a vanilla spiking transformer.
> |Model Type | R1@0.3 |   R1@0.5 |   R1@0.7 |  mAP@avg |  mAP |  HIT@1 | Activity|
> |-----------------------------|---------|---------|---------|---------|-------|-------|----------|
> |Vanilla Spiking transformer |  78.65 |  65.10 |  47.46 |  42.56 |  40.60 |  67.42 | 0.41|
> |SpikingVTG with SFG | 80.72 |  67.42 |  50.65 |  43.81 |  40.74 |  68.32 |  0.34 |
>
> As noted, the SFG mechanism not only leads to a significant performance improvement but also contributes to a reduction in neuronal activity within the network resulting in better computational efficiency.
>
> **B. Effect of Softmax and Layer Normalization**
> |Model Type |  R1@0.3 |   R1@0.5 |   R1@0.7 |  mAP@avg |  @mAP |  HIT@1 | Activity|
> |-----------------------------|---------|---------|---------|---------|-------|-------|----------|
> |SpikingVTG with Softmax & Norm. | 80.72 |  67.42 |  50.65 |  43.81 |  40.74 |  68.32 |  0.34 |
> |SpikingVTG without Softmax & Norm. | 79.87| 66.45| 48.27| 42.68| 40.54 |67.61| 0.25|
>
> As observed there is a slight reduction in model performance w.r.t evaluation metrics however the overall neural activity is reduced resulting in better energy efficiency. Furthermore, since this method does not use any normalization it is better suited for on-chip deployment.

---

> ### Author Response · Authors · 2024-11-15
>
> **C. Effect of Knowledge Distillation**
> |Model Type |  R1@0.3 |  R1 @0.5 |  R1 @0.7 |  mAP@avg |  @mAP |  HIT@1|
> |-----------------------------|---------|---------|---------|---------|-------|-------|
> |SpikingVTG without KD | 67.68 | 52.71 | 34.26 | 32.12 | 35.91 | 57.94 |
> |SpikingVTG with KD | 80.72 |  67.42 |  50.65 |  43.81 |  40.74 |  68.32|
>
> We have added this result to the paper. As observed, KD considerably improves the performance of the model.
>
> **D.Comparison With BPTT**
>
> We did not compare our model with BPTT due to the significantly higher memory requirements of BPTT during training. For instance, training our SpikingVTG model on the QVHighlights dataset requires 25GB of memory with a batch size of 32 and 50 time steps for convergence (T). In contrast, the estimated memory requirement for training with BPTT exceeds 100GB, as BPTT necessitates storing the entire computational graph throughout the training process.Thus, it is infeasible to train the model using BPTT. This also highlights the advantage of the  equilibrium based training mechanism leveraged in this paper. We have added these details in Appendix D.1 titled Comparing implicit differentiation at Equilibrium with BPTT.
>
> **Thank you for your review and comments. In light of the revisions that we have now made in response to your comments, we kindly request that you reconsider your rating. We are happy to address any further feedback you may have.**

---

> > ### Comment · Reviewer_3weo · 2024-11-20
> > **Rebuttal Feedback**
> >
> > I sincerely appreciate the authors' efforts in conducting additional ablation studies to address my questions. However, after carefully reviewing your feedback along with the comments from other reviewers, I believe that some of my concerns, as well as those raised by others, remain insufficiently addressed. Therefore, I would like to adjust my score to align more closely with the assessments of other reviewers (from 3 to 5). Below are the specific reasons for my decision, and I welcome any further discussion on these points:
> >
> > 1. While your model demonstrates marginal improvements over the ANN baseline (UniVTG+PT) on some metrics, it performs worse on others (as shown in Table 1). This raises questions about whether the performance gains can be genuinely attributed to the SNN design and its choices.
> > 2. The primary claim of top performance appears to result from model distillation using an equally top-performing pretrained teacher model. This leaves me questioning whether the claimed contribution should be attributed to the proposed SNN or the effectiveness of the distillation process itself.

---

> > > ### Author Response · Authors · 2024-11-20
> > >
> > > We sincerely appreciate your valuable feedback and your decision to increase the score. I might be missing an update but I am not sure the score has changed yet. Meanwhile, we are actively working on addressing the remaining comments and providing additional experimental results to further address your concerns.

---

> > > > ### Author Response · Authors · 2024-11-22
> > > >
> > > > Thank you for your thoughtful feedback in response to our rebuttal. In this comment, we have addressed your points and also highlighted additional experiments we conducted based on other reviewers' suggestions.
> > > >
> > > > **Comment 1:**
> > > > >While your model demonstrates marginal improvements over the ANN baseline (UniVTG+PT) on some metrics, it performs worse on others (as shown in Table 1). This raises questions about whether the performance gains can be genuinely attributed to the SNN design and its choices.
> > > >
> > > > **Response:**
> > > > Thank you for this insightful question. We selected the pretrained UniVTG as our ANN baseline because it represents the current state-of-the-art in ANN-based VTG architectures. Our goal was to explore the possibility of achieving similar performance within the spiking domain, thereby establishing the first baseline SNN framework for multi-modal VTG tasks.
> > > >
> > > > While we used the pretrained UniVTG model as the teacher during knowledge distillation, our model was only trained on the target dataset (e.g., QVHighlights, Charades, etc.). This means that the knowledge distillation process relied solely on the target dataset, whereas the pretrained UniVTG had been trained on a broader set of datasets, including Ego4D, VideoCC, among others. As a result, during knowledge distillation, we transferred knowledge from the ANN to the SNN for just the target dataset, which led to slightly lower performance on some metrics compared to the teacher model, which was trained more comprehensively. Using our model as a baseline in SNN solutions for VTG, further investigation or extended training on the full set of datasets used in the pretraining of UniVTG could potentially lead to even better performance.
> > > >
> > > >
> > > > We would like to emphasize the primary contribution of this work.
> > > >
> > > > (a) We propose the Saliency Feedback Gating (SFG) mechanism which leverages the temporal dynamics of SNN architectures to perform a multiplicative gating mechanism on the sequence of clips in the video, resulting not only in improved performance over a vanilla spiking transformer but also considerably improving computational efficiency by reducing model-wide neuronal activity (4.3 Ablation Study).  Visualization of SFG mechanism is also added in Appendix D.1.
> > > >
> > > > (b) We demonstrate that incorporating the SFG mechanism preserves the model's convergence dynamics (Figure 3). This stable convergence to equilibrium enables us to leverage implicit differentiation at equilibrium as a robust training framework. Moreover, it facilitates the knowledge distillation from the intermediate states of a 'teacher' ANN to the converged intermediate states of a 'student' spiking model, as discussed in Section 3.4.1. While implicit differentiation at equilibrium has been previously explored, its application was largely restricted to simpler datasets like CIFAR-10/100. For the first time, we scale this approach to multimodal transformer-based SNNs, empowered by the SFG gating mechanism, and empirically demonstrate the convergence dynamics of the underlying model (Figure 3).
> > > >
> > > > (c) This paper introduces, for the first time, a practically viable multi-modal spiking neural network (SNN) architecture that eliminates non-local normalization operations and can undergo extreme weight quantization. Non-local operations like layer normalization and softmax hinder the deployment of many spiking architectures on neuromorphic chips. As detailed in Section 3.4, these model optimizations are achievable only through our multi-staged training pipeline, which integrates an equilibrium-based knowledge distillation (KD) mechanism. Below, we present results comparing a 1-bit, norm-free SpikingVTG model trained from scratch with a variant trained using our proposed multi-staged pipeline. These comparisons highlight the non-trivial nature of the training process introduced in our work.
> > > >
> > > >
> > > > | Model Type                           | R1@0.3  | R1@0.5  | R1@0.7  | mAP@avg | @mAP   | HIT@1  |
> > > > |--------------------------------------|---------|---------|---------|---------|--------|--------|
> > > > | 1-bit NF-SpikingVTG from Scratch     | 66.71   | 50.16   | 32.17   | 31.34   | 34.13  | 56.21  |
> > > > | 1-bit NF-SpikingVTG Multi-Staged pipeline | 78.77   | 65.16   | 47.35   | 42.32   | 40.31  | 67.29  |

---

> ### Author Response · Authors · 2024-11-22
>
> **Comment 2:**
>
> > The primary claim of top performance appears to result from model distillation using an equally top-performing pretrained teacher model. This leaves me questioning whether the claimed contribution should be attributed to the proposed SNN or the effectiveness of the distillation process itself.
>
>
> **Response:**
> Thank you for the question. Below, we present a performance comparison of our model after applying knowledge distillation against two variants of a Spiking Transformer: one without our SFG mechanism and one incorporating it.
>
> | Model Type            | R1@0.3 | R1@0.5 | R1@0.7 | mAP@avg | mAP   | HIT@1  | Activity |
> |-----------------------|--------|--------|--------|---------|-------|--------|----------|
> | model without SFG     | 78.65  | 65.10  | 47.46  | 42.56   | 40.60 | 67.42  | 0.41     |
> | model with SFG        | 80.72  | 67.42  | 50.65  | 43.81   | 40.74 | 68.32  | 0.34     |
>
>
> Both models undergo knowledge distillation followed by fine-tuning. However, the model with the SFG mechanism significantly outperforms the one without it. Additionally, we observe a reduction in neural activity from 0.41 to 0.34, indicating that SFG enhances computational efficiency by promoting sparser spiking patterns. Visualization of SFG mechanism is also added in Appendix D.1.
>
> Moreover, in this paper we for the first-time demonstrate the layer-wise convergence dynamics (convergence to equilibrium states) of the multimodal Spiking VTG architecture (Fig. 3) enabled with the SFG gating mechanism. The ANN-SNN knowledge distillation explored here is made possible after validating this convergence dynamics  of the Spiking VTG model, as shown in Fig. 3 and Eqn. 7. This is because the ANN-SNN knowledge distillation process relies on the converged intermediate states of the Spiking VTG model, making it crucial for it to achieve equilibrium for an efficient and effective KD design. Therefore, another key contribution of our work is that the proposed multi-modal spikingVTG framework (with SFG) facilitates and enables ANN-to-SNN knowledge distillation by leveraging the converged equilibrium states (Fig. 3).
>
> We also evaluated our model on two more dataset, viz TACoS for moment retrieval and Youtube HL for highlight detection. We added the results below for your reference.
>
> Results on TaCOS:
> |Model Type | R1@0.3 |   R1@0.5 |   R1@0.7 |  mIoU |
> |-----------------------------|---------|---------|---------|---------|
> |2D TAN |  40.01 | 27.99 | 12.92 | 27.22 |
> |VSLNet |  35.54 | 23.54 | 13.15 | 24.99 |
> |MDETR |  37.97 | 24.67 | 11.97 | 25.49 |
> |UniVTG | 51.44 | 34.97 | 17.35 | 33.60 |
> |SpikingVTG | 54.32 |  39.16 |  21.78 |  35.78 |
>
> Results on YoutubeHL:
>
> | Model Type   | Dog   | Gym.  | Skating | Skiing | Avg   |
> |--------------|-------|-------|---------|--------|-------|
> | QD-DETR      | 72.2  | 77.4  | 72.7    | 72.8   | 74.4  |
> | UniVTG       | 71.8  | 76.5  | 73.3    | 73.2   | 75.2  |
> | MINI-Net     | 58.2  | 61.7  | 72.2    | 58.7   | 64.4  |
> | Joint-VA     | 64.5  | 71.9  | 62.0    | 73.2   | 71.8  |
> | UMT          | 65.9  | 75.2  | 71.8    | 72.3   | 74.9  |
> | SpikeMba     | 74.4  | 75.4  | 74.3    | 75.5   | 75.5  |
> | SpikingVTG   | 73.90 | 78.07 | 80.10   | 74.20  | 76.55 |
>
> **We sincerely hope that this clarification addresses your concerns, and we would greatly appreciate it if it helps in reconsidering your score. We are more than happy to address any further feedback you may have.**

---

> > ### Author Response · Authors · 2024-11-25
> >
> > Thank you for your insightful and constructive feedback. We have revised our paper to incorporate your suggestions. With the discussion period nearing its conclusion, we kindly request you to reconsider your rating in light of these improvements. Should you have any further feedback, we would be delighted to address it.

---

> > > ### Comment · Reviewer_3weo · 2024-11-26
> > >
> > > Thanks to the author for addressing all my questions and running the extra experiments. Including these experiments and clarifications in the revised manuscript would significantly enhance its presentation and soundness. Based on the improvements, I have decided to increase the score to 6.

---

> > > > ### Author Response · Authors · 2024-11-27
> > > >
> > > > We sincerely appreciate your decision to raise the score and are truly grateful for the constructive feedback that has guided us in improving our work. We have revised our paper to address your suggestions.

---

### Official Review · Reviewer_bDaU · 2024-11-04

**Soundness:** 3
**Presentation:** 2
**Contribution:** 2
**Rating:** 5
**Confidence:** 3

**Summary:**

This manuscript describes an SNN-based model for implementing Video Temporal Grounding. It proposes a baseline structure/model for an SNN transformer suitable for the task, which the authors claim is computationally expensive, and they consider alternatives more computationally sustainable: One based on model distillation from a DNN, then two further variants, one which is simplified by being rid of normalization layers, and another which is further quantized for hardware friendliness/efficiency.

**Strengths:**

In order of importance as I perceived it

A saliency feeback gating (SFG) module is proposed operating somewhat similarly (namely similar complexity) to a transformer block that aims to gate the processing of low saliency parts of the input video stream, thereby reducing computation. This appears to be a novel contribution that did not exist in previous so far literature on VLMs (according to the author's claim).

A version of SNN-based VTG model that attains SoA performance.

**Weaknesses:**

Overall I found the paper a bit cryptic and laconic in its explanations. I think it can be improved by providing a bit more detail about the model structure and implementation of the system, as well as clarifying several ambiguous parts (see suggestions and questions below).

Several of the novelty factors (such as the removal of the normalization layers, or the use of ReLU instead of SoftMax) lack explanation, justification or intuition. Next to that the fact that there are not many datasets available to benchmark with, leaves one with the question whether these "optimizations" work in general (as opposed to being dataset specific).

The main top-performance claim comes from model distillation with an equally top-performance pretrained teacher model, which leaves me with a question whether the actual such claimed-contribution is to be claimed by the authors.

My assessment takes these factors into account, and remains a bit conservative given my limited background in VTG SoA work, but would be happy to elevate the score, upon constructive defense of the paper.

**Questions:**

It is not clear how distances [d_s, d_e] are representing, and therefore also how they are computed. Given that the clips of a video have finite length, should I assume that they measure #clips from the current clip that need to be marked f=1 and scored ? It is therefore also ambiguous for understanding what b_i represents.

Also not clear is if the fixed-length clips of a vide have overlapping segments or not?

What is 1-dimensional Non-Maximum Suppression?

You never introduced f~, s~, b~ in the main text.

u[t+delta] does not make much sense to me. If time is discrete, then delta can only take value in {0,1}, and therefore coincides with u[t+1]. I guess what you intend is to introduce an intermediate variable for Vmem before firing, but I don't comprehend why you relate it to time t, as opposed to using a separate name.

The claim that using spikes (even ternary) enhances -- by definition -- performance is superficial. It is not the case in BNNs, why should it be in SNNs ? And if you refer to efficiency, why is that when there is additionally state to maintain in memory?. As far as I understand the reason SNNs are considered efficient is not grounded on performance and not in absolute terms.

Near lines l-252,253 I believe you need to fix the dimensions or transposes. As they stand, it does not add up correctly.

If I look at equations in (4), i have the impression that there are more than O(L_v D) FP ops. More like O(DL_v + D^2 + L_v^2), which means the size of D vs L_v matters.

From the description I cannot really imagine how the decoder looks like. Pls provide a figure and more explanation(s). Also pls explain how s score is computed with an equation (so far I get the impression that it is the same as a*_i but for the output layer ?)

Around l-308, what is the relaction of act_i[t] with a*_i or a_i ?

It is also not clear if the training in section 3.3 is part of the contribution or other work in literature. To be honest if has been assumed (by me) but not been clear in the paper text how the training work. Namely I assume that you let the network reverberate for a set number of timesteps (T) and then you average the spikes across the timesteps before you use that as a signal to back-propagate. Am I correct?

It is not clear to me the rationale for removing the layer normalization, nor that it would work in all occasions (with different datasets). Unless it has been something that you discovered through trial and error could you elaborate more on the intuition and explanation?

How is ReLU meant to replace softmax? Softmax has many inputs and one output, while ReLU has 1 input and one output. Moreover as with the normalization what is the intuition for removing a probabilistic relative-weighting score and replacing it with an unbounded function that is based on absolute values? And how does that help stabilizing the model.

In l-399 you remark a scaling constant which is not clear to me how it comes to play and why is it used ?

While I find nice and impressive the performance of the SNN, the fact that it comes after distillation of a pretrained DNN, it make me wonder if the performance does belong to the SNN (and its design choices). So first of all the numbers in table 3 come after re-distilling each of those models or not? I would hope each of these models is distilled separately. Then in table 2 the comparison to the other DNNs other than the UniVTG are I would say irrelevant unless you re-distil spiking VTG after each of them. Can you do that for a few of them? (namely test different teachers).

Near l-483 you state that the metric  represents the proportion og active neurons per tstep averaged across all layers. Are they accumulated across timesteps too or ?

In the analysis of energy and power efficiency, first of all you do not discuss power. Second you are considering only encoder attention and linear layers, but not the decoder. Why ? Moreover it is not clear to me if in the MAC/ACC costs you only consider the arithmetic operation or you also take into account the memory transactions to fetch weights, and if you also consider the cost of the memory transactions for the neuron states (Vmem), which is special in the stateful SNN neurons. Can you elaborate more on that say in connection to BNNs which are also quantized and binary but have no (recurrent) state?

---

> ### Author Response · Authors · 2024-11-23
>
> Thank you for your detailed and valuable feedback on our paper. In this rebuttal, we address your comments below.
>
>
> **Comment 1:**
> >Several of the novelty factors (such as the removal of the normalization layers, or the use of ReLU instead of SoftMax) lack explanation, justification or intuition. Next to that the fact that there are not many datasets available to benchmark with, leaves one with the question whether these "optimizations" work in general (as opposed to being dataset specific).
>
> **Response:**
> Thank you for this comment. To further substantiate our contribution. We have evaluated our model on two more datasets namely, TaCOS for moment retrieval tasks and Youtube HL for highlight detection tasks.
>
> Results on TaCOS:
> |Model Type | R1@0.3 |   R1@0.5 |   R1@0.7 |  mIoU |
> |-----------------------------|---------|---------|---------|---------|
> |2D TAN |  40.01 | 27.99 | 12.92 | 27.22 |
> |VSLNet |  35.54 | 23.54 | 13.15 | 24.99 |
> |MDETR |  37.97 | 24.67 | 11.97 | 25.49 |
> |UniVTG | 51.44 | 34.97 | 17.35 | 33.60 |
> |SpikingVTG | 54.32 |  39.16 |  21.78 |  35.78 |
>
> Results on Youtube Highlights (mAP):
>
> | Model Type   | Dog  | Gym. | Skating | Skiing | Avg   |
> |--------------|------|------|---------|--------|-------|
> | QD-DETR      | 72.2 | 77.4 | 72.7    | 72.8   | 74.4  |
> | UniVTG       | 71.8 | 76.5 | 73.3    | 73.2   | 75.2  |
> | MINI-Net     | 58.2 | 61.7 | 72.2    | 58.7   | 64.4  |
> | Joint-VA     | 64.5 | 71.9 | 62.0    | 73.2   | 71.8  |
> | UMT          | 65.9 | 75.2 | 71.8    | 72.3   | 74.9  |
> | SpikeMba     | 74.4 | 75.4 | 74.3    | 75.5   | 75.5  |
> | SpikingVTG   | 73.9 | 78.1 | 80.1    | 74.2   | 76.6  |
>
>
> Our primary motivation for removing layer normalization and softmax operations was to eliminate non-local computations, which pose significant challenges for implementing SNNs on neuromorphic hardware.
>
> Intuition for Replacing Softmax: One of the main reasons why softmax is used in attention mechanism [1] it to normalize the attention score, i.e. it makes all the scores positive and normalize it within a specific range. Our proposed approach of using relu + scaling with {L^-1} maintains this normalization effect without performing nonlocal operations. Empirically we also demonstrate that using our multi-stage training pipeline we are able to achieve similar performance as traditional softmax.
>
> Intuition for Removing Layernorm: As described in Section 3.4, before removing layer normalization, we perform knowledge distillation (KD) on a SpikingVTG model that includes layer normalization. This step stabilizes the initial training by mitigating issues like vanishing or exploding gradients and ensuring effective gradient flow. Through this process, the model learns robust feature representations and meaningful weight configurations. Once layer normalization is removed, fine-tuning the pre-trained model allows it to adapt effectively to operate without normalization. This adaptation has been empirically validated in our experiments.
>
> As detailed in Section 3.4, these model optimizations are achievable only through our multi-staged training pipeline, which integrates an equilibrium-based knowledge distillation (KD) mechanism. Below, we present results comparing a 1-bit, norm-free SpikingVTG model trained from scratch with a variant trained using our proposed multi-staged pipeline. These comparisons highlight the non-trivial nature of the training process introduced in our work.
>
>
> | Model Type                                | R1@0.3 | R1@0.5 | R1@0.7 | mAP@avg | mAP  | HIT@1  |
> |-------------------------------------------|--------|--------|--------|---------|-------|--------|
> | 1-bit NF-SpikingVTG from Scratch          | 66.71  | 50.16  | 32.17  | 31.34   | 34.13 | 56.21  |
> | 1-bit NF-SpikingVTG Multi-Staged pipeline | 78.77  | 65.16  | 47.35  | 42.32   | 40.31 | 67.29  |
>
> Reference
>
> [1] Katharopoulos, Angelos, Apoorv Vyas, Nikolaos Pappas, and François Fleuret. "Transformers are rnns: Fast autoregressive transformers with linear attention." In International conference on machine learning, pp. 5156-5165. PMLR, 2020.

---

> ### Author Response · Authors · 2024-11-23
>
> **Comment 2:**
> >The main top-performance claim comes from model distillation with an equally top-performance pretrained teacher model, which leaves me with a question whether the actual such claimed-contribution is to be claimed by the authors.
>
> **Response:**
> Thank you for the question. Below, we present a performance comparison of our model after applying knowledge distillation against two variants of a Spiking Transformer: one without our SFG mechanism and one incorporating it.
> | Model Type          | R1@0.3 | R1@0.5 | R1@0.7 | mAP@avg | mAP   | HIT@1 | Activity |
> |---------------------|--------|--------|--------|---------|-------|-------|----------|
> | Model without SFG   | 78.65  | 65.10  | 47.46  | 42.56   | 40.60 | 67.42 | 0.41     |
> | Model with SFG      | 80.72  | 67.42  | 50.65  | 43.81   | 40.74 | 68.32 | 0.34     |
>
>
> Both models undergo knowledge distillation followed by fine-tuning. However, the model with the SFG mechanism significantly outperforms the one without it. Additionally, we observe a reduction in neural activity from 0.41 to 0.34, indicating that SFG enhances computational efficiency by promoting sparser spiking patterns. Visualization of SFG mechnaism is also added in Appendix D.1.
>
> Moreover, in this paper we for the first-time demonstrate the layer-wise convergence dynamics (convergence to equilibrium states) of the multimodal Spiking VTG architecture (Fig. 3) enabled with the SFG gating mechanism.. The ANN-SNN knowledge distillation explored here is made possible after validating this convergence dynamics  of the Spiking VTG model, as shown in Fig. 3 and Eqn. 7. This is because the ANN-SNN knowledge distillation process relies on the converged intermediate states of the Spiking VTG model, making it crucial to achieve equilibrium for an efficient and effective KD design. Therefore, another key contribution of our work is that the proposed multi-modal spikingVTG framework facilitates and enables ANN-to-SNN knowledge distillation by leveraging the converged equilibrium states (Fig. 3).
>
>  **Comment 3:**
> >It is not clear how distances [d_s, d_e] are representing, and therefore also how they are computed. Given that the clips of a video have finite length, should I assume that they measure #clips from the current clip that need to be marked f=1 and scored ? It is therefore also ambiguous for understanding what b_i represents.
>
>
> **Response:**
> Thank you for your comment. Each clip in the sequence is centered around a timestamp $t_i$. The offset $d_s$ represents the temporal distance from $t_i$ to the start of the interval, so the start of the boundary is at $t_i - d_s$. Similarly, $d_e$ is the temporal distance to the end of the interval, meaning the end of the boundary is at $t_i + d_e$. Therefore, the boundary parameter $b_i$, which marks the start and end of the interval, is given by $[(t_i - d_s), (t_i + d_e)]$.
>
>
>  **Comment 4:**
> >Also not clear is if the fixed-length clips of a video have overlapping segments or not?
>
> **Response:**
> The fixed-length clips of the video are non-overlapping.
>
>
>  **Comment 5:**
> >What is 1-dimensional Non-Maximum Suppression?
>
> **Response:**
> Due to the high number of predicted boundaries ($L_v$), we apply 1D Non-Maximum Suppression (NMS) with a threshold of 0.7 to remove excessively overlapping bounding boxes, producing the final prediction. If there are two candidates whose temporal IoU is > 0.7, we only keep the candidate with higher $f_i$ value and remove the other one.
>
>  **Comment 6:**
> >You never introduced f, s, b~ in the main text.
>
> **Response:**
> Thank you for highlighting this aspect. We have revised our paper accordingly.
>
>  **Comment 7:**
> >u[t+delta] does not make much sense to me. If time is discrete, then delta can only take value in {0,1}, and therefore coincides with u[t+1]. I guess what you intend is to introduce an intermediate variable for Vmem before firing, but I don't comprehend why you relate it to time t, as opposed to using a separate name.
>
> **Response:**
> Thank you for this comment. Yes, as you correctly pointed out, the $u[t+ \delta]$, simply correspond to an intermediate state of the variable.

---

> > ### Author Response · Authors · 2024-11-23
> >
> > **Comment 8:**
> > >The claim that using spikes (even ternary) enhances -- by definition -- performance is superficial. It is not the case in BNNs, why should it be in SNNs ? And if you refer to efficiency, why is that when there is additionally state to maintain in memory?. As far as I understand the reason SNNs are considered efficient is not grounded on performance and not in absolute terms.
> >
> > **Response:**
> > Thank you his insightful question. In our experiments we saw that ternary spikes achieved slightly better performance over binary spikes. Furthermore, since ternary spike does not introduce floating-point matrix multiplications in our computation we decided to use it for all our models. Below are the results evaluated on QVHighlights dataset.
> > |Model Type |  R1@0.3 |   R1@0.5 |   R1@0.7 |  mAP@avg |  @mAP |  HIT@1 |
> > |-----------------------------|---------|---------|---------|---------|-------|-------|
> > |SpikingVTG with binary spikes | 80.19 | 66.97| 50.13| 43.24| 40.59 |67.56|
> > |SpikingVTG with ternary spikes | 80.72 |  67.42 |  50.65 |  43.81 |  40.74 |  68.32 |
> >
> >
> >  **Comment 9:**
> > >Near lines l-252,253 I believe you need to fix the dimensions or transposes. As they stand, it does not add up correctly.
> >
> > **Response:**
> > Thank you for highlighting this point. We have revised our paper accordingly.
> >
> >  **Comment 10:**
> > >If I look at equations in (4), i have the impression that there are more than O(L_v D) FP ops. More like O(DL_v + D^2 + L_v^2), which means the size of D vs L_v matters.
> >
> > **Response:**
> > Thank you for your comment. In Eqn. 4, the computation of $F_s^{v_i}$ for each clip has a computational cost of $O(D)$, since both vectors involved are $D$-dimensional. To compute $F_s^{v_i}$ for all $L_v$ clips, the total complexity is $O(L_v \cdot D)$. Additionally, the operation $V \cdot F_s^v$ involves element-wise multiplication, which has a complexity of $O(L_v \cdot D)$.
> >
> >
> >  **Comment 11:**
> > >From the description I cannot really imagine how the decoder looks like. Pls provide a figure and more explanation(s). Also pls explain how s score is computed with an equation (so far I get the impression that it is the same as a*_i but for the output layer ?)
> >
> > **Response:**
> > Thank you for highlighting this aspect. We have revised the paper with a figure of the spiking decoder module in Appendix A.2.
> >
> >  **Comment 12:**
> > >Around l-308, what is the relation of act_i[t] with a*_i or a_i ?
> >
> > **Response:**
> > Act_i[t] =  measure the total activity of the neuron at time step t. Since our model supports ternary spikes i.e. both +1 and -1, act_i[t] simply does an average of the spiking events over time, thus it is strictly positive and measure the neuronal activity. $a_i$ on the other hand can be negative as well since it is given by Eqn. 3 (where $s_i$ at any time step can be -1 as well)
> >
> >
> >  **Comment 13:**
> > >It is also not clear if the training in section 3.3 is part of the contribution or other work in literature. To be honest if has been assumed (by me) but not been clear in the paper text how the training work. Namely I assume that you let the network reverberate for a set number of timesteps (T) and then you average the spikes across the timesteps before you use that as a signal to back-propagate. Am I correct?
> >
> > **Response:**
> > Thank you for this insightful comment. In this work, we present for the first time a method where a multi-modal transformer-based SNN can be trained using implicit differentiation at equilibrium. This approach takes advantage of the convergence dynamics of the underlying model (Fig. 3). As the reviewer correctly pointed out, we allow the SNN to reach an equilibrium state, and then use the layer-wise average spiking rate (ASR) (Eqn. 5) at equilibrium to train the model with a single backpropagation step, bypassing the need for the computationally expensive BPTT.

---

> ### Author Response · Authors · 2024-11-23
>
> **Comment 14:**
> >It is not clear to me the rationale for removing the layer normalization, nor that it would work in all occasions (with different datasets). Unless it has been something that you discovered through trial and error could you elaborate more on the intuition and explanation?
>
> **Response:**
> Thank you for this insightful observation. We have addressed this comment in the response to Comment 1.
>
>
>  **Comment 15:**
> >How is ReLU meant to replace softmax? Softmax has many inputs and one output, while ReLU has 1 input and one output. Moreover as with the normalization what is the intuition for removing a probabilistic relative-weighting score and replacing it with an unbounded function that is based on absolute values? And how does that help stabilizing the model.
>
> **Response:**
> Thank you for this insightful observation. We have addressed this comment in the response to Comment 1.
>
>  **Comment 16:**
> >In l-399 you remark a scaling constant which is not clear to me how it comes to play and why is it used ?
>
> **Response:**
> Thank you for this question. The output of the 1-bit quantized linear layer is multiplied by the scaling factor ($\beta = \frac{1}{nm} \sum_{ij} |W_{ij}|$) . This scaling factor helps stabilize the learning process by reducing the L2 error between the real-valued and binarized weights.
>
>
>  **Comment 17:**
> >While I find nice and impressive the performance of the SNN, the fact that it comes after distillation of a pretrained DNN, it make me wonder if the performance does belong to the SNN (and its design choices). So first of all the numbers in table 3 come after re-distilling each of those models or not? I would hope each of these models is distilled separately. Then in table 2 the comparison to the other DNNs other than the UniVTG are I would say irrelevant unless you re-distil spiking VTG after each of them. Can you do that for a few of them? (namely test different teachers).
>
> **Response:**
> Thank you for this comment. We have addressed this comment in the response to Comment 2.
>
>  **Comment 18:**
> >Near l-483 you state that the metric represents the proportion of active neurons per tstep averaged across all layers. Are they accumulated across timesteps too or ?
>
> **Response:**
> Thank you for this question. The metric represents the proportion of active neurons averaged over the total number of time steps (T). In our ablation study, we observed that the 1-bit NF-SpikingVTG exhibits the lowest level of neural activity, which correlates with its superior energy efficiency (Fig. 5).
>
>  **Comment 19:**
> >In the analysis of energy and power efficiency, first of all you do not discuss power. Second you are considering only encoder attention and linear layers, but not the decoder. Why ? Moreover it is not clear to me if in the MAC/ACC costs you only consider the arithmetic operation or you also take into account the memory transactions to fetch weights, and if you also consider the cost of the memory transactions for the neuron states (Vmem), which is special in the stateful SNN neurons. Can you elaborate more on that say in connection to BNNs which are also quantized and binary but have no (recurrent) state?
>
> **Response:**
> Thank you for highlighting this aspect. In the revised paper the discussion on power efficiency is added (lines 520-528). Given that the transformer core represents the most computationally intensive component, we concentrated our energy analysis on this part. However, since our spiking decoder (Appendix A.2) employs the same spike-based computation design, it inherits the same efficiency benefits. We primarily consider the arithmetic operations for doing the analysis [1]. Binary Neural Networks (BNNs) are closely related to Spiking Neural Networks (SNNs), with research [2] demonstrating that binary SNNs can effectively leverage 'In-Memory' hardware accelerators designed specifically for BNNs.
>
>
> References:
>
> [1] Song Han, Jeff Pool, John Tran, and William J. Dally. Learning both weights and connections for
> efficient neural networks, 2015.
> [2] Lu, Sen, and Abhronil Sengupta. "Exploring the connection between binary and spiking neural networks." Frontiers in neuroscience 14 (2020): 535.
>
> **Thank you for your review and comments. In light of the revisions that we have now made in response to your comments, we kindly request that you reconsider your rating. We are happy to address any further feedback you may have.**

---

> ### Comment · Reviewer_bDaU · 2024-11-25
>
> Thank you for all the clarifications. Below are some additional comments.
>
> > Response: Thank you for this comment. To further substantiate our contribution. We have evaluated our model on two more datasets namely, TaCOS for moment retrieval tasks and Youtube HL for highlight detection tasks.
>
> OK but if you distilled your models with the pre-trained UniVTG (UniVTG+PT), why do you not include them also in the tables? As I mentioned in my original comments to me this is the most informative row for comparison, alongside your method without SFG and your method with SFG. Also why do you not add these extra tests in the paper? (appendix if they do not fit in the main text page budget).
>
> > Intuition for Replacing Softmax: One of the main reasons why softmax is used in attention mechanism [1] it to normalize the attention score, i.e. it makes all the scores positive and normalize it within a specific range. Our proposed approach of using relu + scaling with {L^-1} maintains this normalization effect without performing nonlocal operations. Empirically we also demonstrate that using our multi-stage training pipeline we are able to achieve similar performance as traditional softmax.
>
> Hmm, is this *your* result/innovation, or the paper that introduced this "surrogate" for SoftMax ?
>
> > As detailed in Section 3.4, these model optimizations are achievable only through our multi-staged training pipeline, which integrates an equilibrium-based knowledge distillation (KD) mechanism. Below, we present results comparing a 1-bit, norm-free SpikingVTG model trained from scratch with a variant trained using our proposed multi-staged pipeline. These comparisons highlight the non-trivial nature of the training process introduced in our work.
>
> Sure i understand that, but at the end of the day you only perform fine-tuning (twice) to recover from the accuracy degradation, which is not something new, right? (it has a long history in the literature of quantization where it serves the same purpose). In all honesty your multistage training framework i don't view it as a contribution since this is a natural measure one would take in the case of lost accuracy due to a change in the network configuration or the input distribution. Having said that I like that you do not loose much accuracy when going down to binary weights.
>
> > Moreover, in this paper we for the first-time demonstrate the layer-wise convergence dynamics (convergence to equilibrium states) of the multimodal Spiking VTG architecture (Fig. 3) enabled with the SFG gating mechanism..
>
> This sentence, I do not understand. You are applying the SFG feedback from the encoder output back to the encode input, right? So what layer-wise dynamics are you referring to ? An SNN layer with mangled input, let to reverberate for a number or timesteps, will eventually converge (subject to spectral properties of the weights). If you are saying that your added SFG feedback loop is forcing the entire network to converge to steady-state dynamics faster, I can believe that (that is the essence of feedback loops, and that is the essence of the FORCE algorithm https://www.nature.com/articles/s41467-017-01827-3). I m not trying to pull your leg, but if that is the key innovation, I would hope your evaluation to focus on this. For example if a query sentence should retrieve 2 or 3 parts (groups of clips) in the video then I would like to see how SFG differs from absence of SFG using plots like the one in D.1, and that it does not focus only on 1 part for instance. Having said that I realise time is tight, now.
>
> > Response: Thank you for highlighting this aspect. In the revised paper the discussion on power efficiency is added (lines 520-528). Given that the transformer core represents the most computationally intensive component, we concentrated our energy analysis on this part. However, since our spiking decoder (Appendix A.2) employs the same spike-based computation design, it inherits the same efficiency benefits. We primarily consider the arithmetic operations for doing the analysis [1]. Binary Neural Networks (BNNs) are closely related to Spiking Neural Networks (SNNs), with research [2] demonstrating that binary SNNs can effectively leverage 'In-Memory' hardware accelerators designed specifically for BNNs.
>
> Fair enough then, but I think it would be also fair to put your assumptions upfront in the text (or at least the appendix). Namely that you are assuming analog in-memory computing (and not digital neuromorphic accelerators like Loihi and TrueNorth that you cite in the text), so that you do not account for cost of memory I/O transactions. In digital neuromorphic accelerators the existence of state and leakage can make a dramatic difference compared to BNNs, and the interactions with the memory is the main drain of energy.

---

> ### Author Response · Authors · 2024-11-25
>
> Thank you for your thoughtful feedback on our rebuttal. In this response, we have carefully addressed the remaining points you raised.
>
> **Comment 1**
>
> >OK but if you distilled your models with the pre-trained UniVTG (UniVTG+PT), why do you not include them also in the tables? As I mentioned in my original comments to me this is the most informative row for comparison, alongside your method without SFG and your method with SFG. Also why do you not add these extra tests in the paper?
>
> **Response:**
> Thank you for this comment. We have added these new experiments in Appendix D.3 of the revised paper and have also added the results of the teacher model. For TaCOS dataset we have also added the result when SFG is not used.
>
> **Comment 2**
> >Hmm, is this your result/innovation, or the paper that introduced this "surrogate" for SoftMax ?
>
> **Response:**
> In this paper, we introduce for the first time the ReLU and $L^{-1}$scaling-based activation in the spiking domain, replacing the traditional softmax. Our multi-modal model utilizes ternary spikes, and through experimental evaluation, we find that the combination of ReLU and scaling by $L^{-1}$ allow us to achieve performance comparable to that of softmax. Additionally, we present, for the first time, the complete removal of non-local layer normalization with minimal impact on model performance. In contrast, all prior works on spiking transformer architectures have relied on normalization as a core operation. To the best of our knowledge, SpikingVTG is also the first multimodal spiking video-language model and 1-bit NF-SpikingVTG is a first-of-its-kind norm. free 1-bit spiking transformer architecture.
>
> **Comment 3**
> >Sure i understand that, but at the end of the day you only perform fine-tuning (twice) to recover from the accuracy degradation, which is not something new, right? (it has a long history in the literature of quantization where it serves the same purpose). In all honesty your multistage training framework i don't view it as a contribution since this is a natural measure one would take in the case of lost accuracy due to a change in the network configuration or the input distribution. Having said that I like that you do not loose much accuracy when going down to binary weights.
>
> **Response:**
> Yes, we agree that fine-tuning techniques are commonly used to recover accuracy degradation, and we acknowledge the long history of this approach in the quantization literature. However, our multi-stage training pipeline not only allows us to develop a Spiking model, but it also demonstrates the feasibility of removing all non-local normalization operations and applying extreme weight quantization to an SNN with minimal performance degradation (Table 3). This process, which combines knowledge distillation and fine-tuning, presents a way to address challenges specific to SNNs, particularly with respect to maintaining performance under these extreme conditions (like 1-bit quantization / removing non-local ops.) which are more relevant in a resource constraint low-powered environment where SNNs are leveraged.

---

> ### Author Response · Authors · 2024-11-25
>
> **Comment 4**
> >This sentence, I do not understand. You are applying the SFG feedback from the encoder output back to the encode input, right? So what layer-wise dynamics are you referring to ? An SNN layer with mangled input, let to reverberate for a number or timesteps, will eventually converge (subject to spectral properties of the weights). If you are saying that your added SFG feedback loop is forcing the entire network to converge to steady-state dynamics faster, I can believe that (that is the essence of feedback loops, and that is the essence of the FORCE algorithm https://www.nature.com/articles/s41467-017-01827-3). I m not trying to pull your leg, but if that is the key innovation, I would hope your evaluation to focus on this. For example if a query sentence should retrieve 2 or 3 parts (groups of clips) in the video then I would like to see how SFG differs from absence of SFG using plots like the one in D.1, and that it does not focus only on 1 part for instance. Having said that I realise time is tight, now.
>
> **Response:**
> Thank you for highlighting this. The SFG mechanism not only improves performance in terms of the evaluation metrics but, as shown in Sec. 4.2 (Table 3), it also leads to a reduction in model-wide neural activity. This phenomenon is further illustrated in the convergence dynamics in Fig. 3b. Therefore, the key advantage of SFG is twofold: it improves performance while also making the model more computationally efficient.
>
> While SNN models typically converge to equilibrium given an input, this paper is the first to demonstrate that this convergence can be leveraged to train not only deep multi-modal SNNs (beyond simple vision-based data), but also model variants that exclude non-local normalizations (such as layer normalization and softmax), as well as those with extremely quantized 1-bit weights. Our 1-bit NF-SpikingVTG variant is not only an unique solution for low-power video temporal grounding but also paves the way for developing similar transformer architectures for a wide range of tasks.
>
>
> **Comment 5**
> >Fair enough then, but I think it would be also fair to put your assumptions upfront in the text (or at least the appendix). Namely that you are assuming analog in-memory computing (and not digital neuromorphic accelerators like Loihi and TrueNorth that you cite in the text), so that you do not account for cost of memory I/O transactions. In digital neuromorphic accelerators the existence of state and leakage can make a dramatic difference compared to BNNs, and the interactions with the memory is the main drain of energy.
>
> **Response:**
> Thank you for your suggestion. We have incorporated it into the revised paper and addressed it in Sec. 4.4.
>
> **Thank you for your constructive feedback. It helped us improve our work**

---

> > ### Author Response · Authors · 2024-11-28
> >
> > Thank you for your insightful and constructive feedback. We have revised our paper to incorporate your suggestions. With the paper revision period nearing its conclusion, we kindly request you to reconsider your rating in light of these improvements. Should you have any further feedback, we would be delighted to address it.

---

> > > ### Author Response · Authors · 2024-11-30
> > >
> > > Thank you for your valuable suggestions and feedback, which we have integrated into our revised paper. As the paper discussion period is drawing to a close, we kindly request that you reconsider your rating based on the updates made. We welcome any additional feedback and are happy to address it.

---

> > > > ### Author Response · Authors · 2024-12-01
> > > >
> > > > Thank you for your thoughtful suggestions. We have incorporated the recommended clarifications and added new experimental results to strengthen our paper. With the discussion period concluding tomorrow, we kindly request you to reconsider your rating in light of these updates. If you have any additional feedback or comments, we would be delighted to address them.

---

> > > > > ### Comment · Reviewer_bDaU · 2024-12-02
> > > > >
> > > > > You have made a significant effort and work to provide additional results and revise the paper, and I fully respect/acknoweledge that.
> > > > >
> > > > > However the additional results presented to me and the other reviewers, mainly reinforce most of my initial impressions.
> > > > > This is for the following reasons:
> > > > >  - the novelty claimed are mainly limited to one imho, the SFG mechanism, the rest is a composition of already existing innovations (relu, distillation, iterative fine-tuning, etc)
> > > > >  - the evaluation is not targeted sufficiently on the SFG mechanism, which would shed light to its merits (e.g. ability to retrieve multiple clips relevant to a request).
> > > > >  - the comparisons to the ANN literature are somewhat superficial as they are based on the proxy score  (through distillation) of a pretrained ANN solution.
> > > > >  - the steps to go from ANN to SNN bears also limited novelty, and the removal of normalization does hurt performance after all even post training (more visible in the new datasets)
> > > > >  - a comparison to BNNs when it comes to energy efficiency which would be an apples-to-apples comparison with similar steps for refinement (since your framework essentially is based on the quantization liteterature), has not been considered.
> > > > >
> > > > > I am hesitant to revise my score at this moment, but I may reconsider in order to align myself with the remaining reviews in the ballot time.

---

> ### Author Response · Authors · 2024-12-02
>
> Thank you for your feedback. We have added a response to your concerns below.
>
> - The SFG mechanism is a key architectural innovation in our work, specifically designed to handle multi-modal video-language input data. It leads to both improved model performance on evaluation metrics and enhanced computational efficiency. As mentioned by the reviewer and highlighted in the introduction of our paper, couple of the optimization techniques—such as replacing softmax or employing extreme quantization—have been explored in conventional non-spiking settings. However, leveraging these techniques in the spiking domain presents significant challenges because of spike-based communication in SNN models. Moreover, our proposed 1-bit NF SpikingVTG model represents the first normalization-free, 1-bit transformer architecture for spiking neural networks (Energy efficiency highlighted in Fig. 5). This approach is particularly relevant for resource-constrained neuromorphic chips and, to the best of our knowledge, has not been explored in prior literature.
>
> - The QVHighlights dataset, used to evaluate our model (Tables 1, 2, and 3), consists of multiple test samples where a single query can correspond to multiple disjoint video clips. Our model performs comparably to state-of-the-art methods across most evaluation metrics on QVHighlights, and in some cases, it even achieves superior performance. In the final version of the paper, we will be happy to give input/output examples of this specific scenario in more detail. The contribution of the SFG mechanism in enhancing both model performance and computational efficiency, compared to the vanilla spiking transformer model, is discussed in Section 4.3.
>
> - We compared our model with other ANN-based models because our work represents the first spiking solution to the VTG task. Additionally, in our ablation study, we evaluate the different variants of the SpikingVTG model proposed in our paper.
>
> - Our model does not rely on a conventional ANN-to-SNN conversion framework. Instead, by using knowledge distillation, we directly train the underlying 'student' SNN model. This approach allows us, for the first time, to scale the training of a multi-modal SNN architecture leveraging the convergence dynamics of the underlying SNN. It's important to note that the SFG mechanism we employ is unique to the 'student' SNN model, which helps explain why our model outperforms the 'teacher' in certain metrics. Additionally, the process of removing layer normalization, replacing softmax, and applying 1-bit quantization to a multi-modal SNN is explored for the first time in this paper. This approach enables the deployment of these complex models on resource-constrained hardware.
>
> - BNNs are indeed highly computationally efficient. However, developing a BNN-based solution for VTG tasks presents a unique and interesting research challenge. Unlike SNNs, BNNs lack the inherent temporal dynamics, and as a result, the 1-bit activation quantization can significantly impact model performance. Since there are no existing BNN solutions for VTG tasks, we chose to compare our approach against the current state-of-the-art non-spiking architectures.

---

### Official Review · Reviewer_J8S5 · 2024-11-04

**Soundness:** 3
**Presentation:** 3
**Contribution:** 2
**Rating:** 6
**Confidence:** 3

**Summary:**

This work introduces the SpikingVTG model for Video Temporal Grounding (VTG) by incorporating spiking neural networks (SNNs). To improve performance and reduce neural activity, this work proposes the saliency feedback gating mechanism (SFG), which also allows training with implicit differentiation at equilibrium to avoid BPTT. To further enhance computation efficiency, a multi-stage training pipeline that utilizes knowledge distillation and architectural adjustments, is used to develop the Normalization Free (NF)-SpikingVTG model and the 1-bit NF-SpikingVTG model. Experiments on QVHighlights and Charades-STA demonstrate the advantages of SpikingVTG in terms of performance and energy efficiency.

**Strengths:**

1. Introducing SNNs to VTG is an interesting exploration, which leads to an energy-saving model for the field.
2. The saliency feedback gating mechanism effectively conveys dynamic information to filter the inputs, which not only leads to sparser neural activity, but also improves model performance. Furthermore, introducing feedback connections allows the model to be trained with implicit differentiation at equilibrium.
3. The multi-stage training pipeline incorporating knowledge distillation and architectural simplification develops models with higher computational efficiency and low degradation in performance.

**Weaknesses:**

Major points:
1. The main weakness is that the network cannot be considered a fully spiking neural network, and it's not clear if it could be deployed on existing neuromorphic chips. Although the computational complexity of SFG is much lower compared to spiking transformers, it is still impossible to ignore the floating-point multiplication operations and nonlinear computations (softmax), which are not in line with the characteristics of SNN. In addition, the gelu in the intermediate layer (line 729) and the floating-point communications through the residual connection in the output layer (Figure 1 and line 735) are also problematic. Therefore, the network is best called a hybrid SNN and I think the authors should change "Yes" to "Partial" in Table 1 and 2.
2. The technical value of this work seems to be mostly an engineering contribution, and I'm not sure about the value of the innovation, since many key methods come from other work, such as training with implicit differentiation at equilibrium [1], replacing softmax with relu [2], and 1-bit quantization [3].
3. Given that the network is a hybrid SNN, it performs only marginally better than SpikeMba, which is also a hybrid SNN, on some metrics, and even worse on others (Table 1). Besides, there is a lack of performance comparison between the two proposed variants and non-spiking models.

Minor points:
1. In addition to comparing performance, it would be better to report the inference time and the number of model parameters to provide more evidence of the computational efficiency.
2. The numbering in section 4.3.1, "Analysis of Energy and Power Efficiency", is incorrect. It should be at the same level as section 4.3, "Ablation Study", but not a subsection of it.
3. Numerous equations are mixed in with the text, making them less prominent. This is particularly evident in the appendix section on the loss function, where the total loss function consists of three separate loss functions. The formulas for the first two loss functions are presented separately, but the formula for the last term $L_c$ is mixed in with the text, which may cause inconvenience when reading.

[1] Mingqing Xiao, Qingyan Meng, Zongpeng Zhang, Yisen Wang, and Zhouchen Lin. Training feedback spiking neural networks by implicit differentiation on the equilibrium state. Advances in Neural Information Processing Systems. 2021.

[2] Kai Shen, Junliang Guo, Xu Tan, Siliang Tang, Rui Wang, and Jiang Bian. A study on relu and softmax in transformer. arXiv. 2023.

[3] Hongyu Wang, Shuming Ma, Li Dong, Shaohan Huang, Huaijie Wang, Lingxiao Ma, Fan Yang, Ruiping Wang, Yi Wu, and Furu Wei. Bitnet: Scaling 1-bit transformers for large language models. arXiv. 2023.

**Questions:**

1. How do the authors remove the layer normalization in section 3.4.2? Is it simply removed or are other adjustments introduced? It would be good to include details of the implementation in the text.

---

> ### Author Response · Authors · 2024-11-22
>
> Thank you for your detailed and valuable feedback on our paper. In this rebuttal, we address your comments below.
>
>  **Comment 1:**
> >The main weakness is that the network cannot be considered a fully spiking neural network, and it's not clear if it could be deployed on existing neuromorphic chips. Although the computational complexity of SFG is much lower compared to spiking transformers, it is still impossible to ignore the floating-point multiplication operations and nonlinear computations (softmax), which are not in line with the characteristics of SNN. In addition, the gelu in the intermediate layer (line 729) and the floating-point communications through the residual connection in the output layer (Figure 1 and line 735) are also problematic. Therefore, the network is best called a hybrid SNN and I think the authors should change "Yes" to "Partial" in Table 1 and 2.
>
> **Response:**
> Thank you for this insightful comment.
>
> (i) In the Norm-Free (NF) SpikingVTG variant (Table 3), we eliminate all softmax and layer normalization operations, including those used for computing M (as defined in Equation 4) within the SFG block. While this detail may not have been explicitly stated in the original version, we have clarified it in the revised version to ensure completeness. Furthermore, as noted by the reviewer, the computational cost of SFG is significantly less than the spiking transformer core. This is because the primary computations involved in the SFG mechanism are element-wise multiplication and addition operations  ( O(L_v * D) ) which are much less costly than floating-point multiplicative and accumulative (MAC) operations in linear layers O(L * D^2) or attention layers O(LD^2 + L^2D). Furthermore, the value of M (Eqn. 4), needs to be computed only once, as it remains constant across all time steps. This allows it to be cached, further reducing computational overhead.
>
> (ii) During actual operation of the model (i.e. at inference time)  all inter- and intra- layer communication are done using spikes. Thus, the residual connections (in Fig 1) also involve spikes and not floating-point values. The Eqn. in line 735, which the reviewer noted, represents the fixed-point dynamics of the model at equilibrium, thus it involves the converged average spiking rate ($a^*_p$) instead of spike.
>
> Following the reviewers suggestion we performed an ablation study by replacing gelu with a hardware friendly **relu** layer [1]. The results are added in the revised paper and are also added below.
>
> |Model Type |  R1@0.3 |   R1@0.5 |   R1@0.7 |  mAP@avg |  mAP |  HIT@1 |
> |-----------------------------|---------|---------|---------|---------|-------|-------|
> |1-bit NF-SpikingVTG with gelu | 78.77 | 65.16 | 47.35 | 42.32 | 40.31 | 67.29 |
> |1-bit NF-SpikingVTG with relu | 78.39 | 66.06 | 47.10 | 41.78 | 40.22 | 67.10 |
>
> This demonstrates that replacing **gelu** with  **relu** activation does not result in significant reduction in performance. With this change the primary computation of the spiking transformer core can be potentially operated on a neuromorphic chip.

---

> ### Author Response · Authors · 2024-11-22
>
> **Comment 2:**
> >The technical value of this work seems to be mostly an engineering contribution, and I'm not sure about the value of the innovation, since many key methods come from other work, such as training with implicit differentiation at equilibrium [1], replacing softmax with relu [2], and 1-bit quantization [3].
>
> **Response:**
> Thank you highlighting this point. We would like to emphasize the primary contribution of this work which goes beyond engineering contribution.
>
> (a) We propose the Saliency Feedback Gating (SFG) mechanism which leverages the temporal dynamics of SNN architectures to perform a multiplicative gating mechanism on the sequence of clips in the video, resulting not only in improved performance over a vanilla spiking transformer but also considerably improving computational efficiency by reducing model-wide neuronal activity (Sec. 4.3 Ablation Study).  Visualization of SFG mechnaism is also added in Appendix D.1.
>
> (b) Through empirical evaluation, we validate that incorporating the SFG mechanism preserves the model's convergence dynamics (Figure 3). This stable convergence to equilibrium enables us to leverage implicit differentiation at equilibrium as a robust training framework. Moreover, it facilitates the knowledge distillation from the intermediate states of a 'teacher' ANN to the converged intermediate states of a 'student' spiking model, as discussed in Section 3.4.1. While implicit differentiation at equilibrium has been previously explored, as noted by the reviewer, its application was largely restricted to simpler datasets like CIFAR-10/100. For the first time, we scale this approach to multimodal transformer-based spiking neural networks, empowered by the SFG gating mechanism, and empirically demonstrate the convergence dynamics of the underlying model (Figure 3).
>
> (c) As noted by the reviewer and emphasized in the Introduction of our work, techniques such as replacing softmax or employing 1-bit quantization have been explored in the non-spiking domain. However, this paper introduces, for the first time, a practically viable multi-modal spiking neural network (SNN) architecture that eliminates non-local normalization operations and can undergo extreme weight quantization. As detailed in Section 3.4, these model optimizations are achievable only through our multi-staged training pipeline, which integrates an equilibrium-based knowledge distillation (KD) mechanism. Below, we present results comparing a 1-bit, norm-free SpikingVTG model trained from scratch with a variant trained using our proposed multi-staged pipeline. These comparisons highlight the non-trivial nature of the training process introduced in our work.
>
> | Model Type                                | R1@0.3 | R1@0.5 | R1@0.7 | mAP@avg | @mAP  | HIT@1 |
> |-------------------------------------------|--------|--------|--------|---------|-------|-------|
> | 1-bit NF-SpikingVTG from Scratch          | 66.71  | 50.16  | 32.17  | 31.34   | 34.13 | 56.21 |
> | 1-bit NF-SpikingVTG Multi-Staged pipeline | 78.77  | 65.16  | 47.35  | 42.32   | 40.31 | 67.29 |
>
>
>  **Comment 3:**
> >Given that the network is a hybrid SNN, it performs only marginally better than SpikeMba, which is also a hybrid SNN, on some metrics, and even worse on others (Table 1). Besides, there is a lack of performance comparison between the two proposed variants and non-spiking models.
>
> **Response:**
> i) Thank you for this comment. The core computational paradigm of SpikeMba is not spiking. Instead, it incorporates a spiking component to generate candidate frames, while its primary computational framework is rooted in the ANN-based Mamba architecture, which uses substantial floating point matrix multiplication operations as well as non-local normalization operations.
> ii) To further substantiate our contribution. We have evaluated our model on two more datasets namely, TaCOS for moment retrieval tasks and Youtube Highlights for highlight detection tasks.
>
> Results on **TACoS**:
>
> | Model Type     | R1@0.3 | R1@0.5 | R1@0.7 | mIoU  |
> |----------------|--------|--------|--------|-------|
> | 2D TAN         | 40.01  | 27.99  | 12.92  | 27.22 |
> | VSLNet         | 35.54  | 23.54  | 13.15  | 24.99 |
> | MDETR          | 37.97  | 24.67  | 11.97  | 25.49 |
> | UniVTG         | 51.44  | 34.97  | 17.35  | 33.60 |
> | SpikingVTG     | 54.32  | 39.16  | 21.78  | 35.78 |
>
>
> Results on **Youtube Highlights** (mAP):
>
> | Model Type   | Dog   | Gym.  | Skating | Skiing | Avg   |
> |--------------|-------|-------|---------|--------|-------|
> | QD-DETR      | 72.2  | 77.4  | 72.7    | 72.8   | 74.4  |
> | UniVTG       | 71.8  | 76.5  | 73.3    | 73.2   | 75.2  |
> | MINI-Net     | 58.2  | 61.7  | 72.2    | 58.7   | 64.4  |
> | Joint-VA     | 64.5  | 71.9  | 62.0    | 73.2   | 71.8  |
> | UMT          | 65.9  | 75.2  | 71.8    | 72.3   | 74.9  |
> | SpikeMba     | 74.4  | 75.4  | 74.3    | 75.5   | 75.5  |
> | SpikingVTG   | 73.90 | 78.07 | 80.10   | 74.20  | 76.55 |

---

> ### Author Response · Authors · 2024-11-22
>
> **Comment 4:**
> >In addition to comparing performance, it would be better to report the inference time and the number of model parameters to provide more evidence of the computational efficiency.
>
> **Response:**
> Thank you for this feedback. Our model consists of 41.8M  parameters (322 MB), which is comparable in scale to all the baselines, including UniVTG and UMT. The memory footprint of our 1-bit variant is significantly reduced (11 MB approx.), as all linear layer weights are stored in 1-bit precision instead of full precision, as used in other variants. On the QVHighlights evaluation dataset, our model achieves an inference time of 45 seconds, with operation time step (T) being 10 and a batch size of 8. Details about the GPU used for these experiments are provided in Section 4.1.
>
>  **Comment 5:**
> >The numbering in section 4.3.1, "Analysis of Energy and Power Efficiency", is incorrect. It should be at the same level as section 4.3, "Ablation Study", but not a subsection of it.
>
> **Response:**
> Thank you for highlighting this. We have fixed it in our paper.
>
>  **Comment 6:**
> >Numerous equations are mixed in with the text, making them less prominent. This is particularly evident in the appendix section on the loss function, where the total loss function consists of three separate loss functions. The formulas for the first two loss functions are presented separately, but the formula for the last term  is mixed in with the text, which may cause inconvenience when reading.
>
> **Response:**
> Thank you for highlighting this aspect. We have updated our paper accordingly.
>
>  **Comment 7:**
> >How do the authors remove the layer normalization in section 3.4.2? Is it simply removed or are other adjustments introduced? It would be good to include details of the implementation in the text.
>
> **Response:**
> Thank you for your insightful comment. Our multi-staged training framework enables the removal of layer normalization without a significant drop in model performance. The process for removing layer normalization is as follows:
>
> Initial Training with Layer Norm: We first train the SpikingVTG model with layer normalization enabled, applying knowledge distillation (KD) and fine-tuning, leveraging the model's convergence dynamics, as illustrated in Figures 3 and 4.
>
> Layer Norm Removal and Fine-Tuning: After this, we remove the layer normalization operations from the trained model and perform an additional round of fine-tuning.
>
> As outlined in Section 3.4.1, our proposed KD method, which utilizes the convergence dynamics of the model (Fig. 3), plays a crucial role in facilitating the removal of layer normalization. This is further demonstrated in the table provided in response to Comment 2.
>
>
> References:
>
> [1] Timcheck, Jonathan, Sumit Bam Shrestha, Daniel Ben Dayan Rubin, Adam Kupryjanow, Garrick Orchard, Lukasz Pindor, Timothy Shea, and Mike Davies. "The Intel neuromorphic DNS challenge." Neuromorphic Computing and Engineering 3, no. 3 (2023): 034005.
>
> **Thank you for your review and comments. In light of the revisions that we have now made in response to your comments, we kindly request that you reconsider your rating. We are happy to address any further feedback you may have.**

---

> ### Author Response · Authors · 2024-11-25
>
> Thank you for your insightful and constructive feedback. We have revised our paper to incorporate your suggestions. With the discussion period nearing its conclusion, we kindly request you to reconsider your rating in light of these improvements. Should you have any further feedback, we would be delighted to address it.

---

> ### Comment · Reviewer_J8S5 · 2024-11-26
>
> Thanks a lot for clarifications. The additional experimental results address my concerns and I decide to raise my score.

---

> > ### Author Response · Authors · 2024-11-27
> >
> > We truly appreciate your decision to increase your score and are deeply grateful for the positive feedback that has helped us improve our work.

---

### Author Response · Authors · 2024-11-24

We sincerely thank all the reviewers for their thoughtful comments and constructive feedback. We have carefully addressed each reviewer's comments in our individual rebuttals and have revised the paper accordingly to incorporate their feedback. As the discussion period is nearing its end, we look forward to engaging further to ensure all feedback and suggestions are thoroughly addressed.

---

### Author Response · Authors · 2024-12-02

We sincerely thank all the reviewers for their valuable comments and suggestions, which have greatly contributed to enhancing our work during the discussion period. While we have carefully addressed each of the reviewers' comments, below we highlight some of the key revisions made to the paper:

(a) In response to the valuable suggestions from reviewers **J8S5**, **bDaU**, **pBmG**, and **zh6n**, we have broadened our evaluation to include two additional datasets (Appendix D.3): TaCOS for moment retrieval tasks and Youtube Highlights for highlight detection tasks, thereby providing a more comprehensive assessment of our model's performance. Moreover, as suggested by reviewer **pBmG**, we have added more baseline models for comparison. We would like to kindly highlight that, as our work is the first spiking solution for the multi-modal VTG task, all the baselines included in the comparison are non-spiking models.

(b) We have added a section (Appendix D.1) visualizing the proposed Saliency Feedback Gating (SFG) mechanism, as recommended by reviewer **pBmG**.

(c) In response to the suggestion from reviewer **J8S5**, we conducted an ablation study to improve the on-chip deployability of our 1-bit NF SpikingVTG model by replacing the GELU activation with the more hardware-friendly ReLU (Table 3). The findings demonstrate that this substitution does not cause a significant performance drop, further highlighting the model's effectiveness for deployment on resource-constrained hardware.

(d) We have added additional ablation studies (Table 3) to substantiate the contributions of our proposed SFG mechanism, multi-stage training pipeline (Norm. Free (NF)-SpikingVTG, 1-bit NF SpikingVTG), as suggested by the reviewers **3weo** and **bDaU**.

**We are grateful for the opportunity to address the reviewers' comments and would like to extend our sincere thanks to all the reviewers for such a constructive and enriching discussion period.**

---

### Meta-Review · Area_Chair_WBKy · 2024-12-20

**Metareview:**

This paper proposed a spiking neural network for video temporal grounding, evaluated on moment retrieval and highlight detection. Five reviewers evaluated the paper, discussed it with the authors, and evolved their ratings during the peer review process. The initial weaknesses raised were

- The proposition is not purely spiking,
- The origin of the claims is potentially in the teacher model,
- Novelty, as most of the building blocks have been taken from existing work,
- Lukewarm performance,
- Missing baselines, and missing ablations,
- Writing and (lack of) clarity.

The authors could address some points, in particular by providing additional experiments, and several ratings were increased during the reviewers/authors discussion. The paper stayed borderline during the review process, and in the reviewers/AC discussions some reviewers expressed their assessment that the paper is not yet ready in spite of having provided a "6 / marginally above the acceptance threshold" rating.

The AC sides with the critical points and judges that the paper is not yet ready for publication. Key to this assessment were the remaining critical weaknesses, which the authors could not address: compared to the state-of-the-art, novelty of this paper is limited to the SFG mechanism, which has not been sufficiently ablated, as pointed out by reviewer bDaU, ie. the main contribution is not backed up by experiments. The evaluation has been judged to be unconvincing. The main performance gains seem to stem from a pre-trained teacher ANN, and the positioning and motivation of the paper provided by the authors seems to have shifted from the initial writing of the paper towards the end of the discussion phase, which left more questions open than it answered.

**Additional Comments On Reviewer Discussion:**

The reviewers engaged with the authors, and discussed the paper with the AC.

---

### Decision · Program_Chairs · 2025-01-22

Reject